# Adversarial Examples for $k$-Nearest Neighbor Classifiers Based on Higher-Order Voronoi Diagrams

**Chawin Sitawarin**
UC Berkeley
chawins@eecs.berkeley.edu

**Evgenios M. Kornaropoulos**
George Mason University
evgenios@gmu.edu

**Dawn Song**
UC Berkeley
dawnsong@cs.berkeley.edu

**David Wagner**
UC Berkeley
daw@cs.berkeley.edu

## Abstract

Adversarial examples are a widely studied phenomenon in machine learning models. While most of the attention has been focused on neural networks, other practical models also suffer from this issue. In this work, we propose an algorithm for evaluating the adversarial robustness of $k$-nearest neighbor classification, i.e., finding a minimum-norm adversarial example. Diverging from previous proposals, we propose the first geometric approach by performing a search that expands outwards from a given input point. On a high level, the search radius expands to the nearby higher-order Voronoi cells until we find a cell that classifies differently from the input point. To scale the algorithm to a large $k$, we introduce approximation steps that find perturbation with smaller norm, compared to the baselines, in a variety of datasets. Furthermore, we analyze the structural properties of a dataset where our approach outperforms the competition.

## 1 Introduction

It is well-known that machine learning models can easily be misled by maliciously crafted inputs, called adversarial examples, which are generated by adding a tiny perturbation to test samples [Biggio et al., 2013, Szegedy et al., 2013, Goodfellow et al., 2015]. Adversarial examples are often studied in the context of neural networks, leaving the problem largely unexplored for other classifiers. The $k$-nearest neighbor, or simply $k$-NN, classifier is a simple yet widely used model in various applications such as data mining, recommendation, and anomaly detection systems where interpretability and simplicity are preferred [Wu et al., 2008]. This non-parametric classifier does not require a training phase and has a well-understood and elegant geometric foundation [Okabe et al., 1992]. $k$-NN is also an active area of research with lots of developments in popular libraries like Google's ScaNN [Guo et al., 2020] and Facebook's FAISS [Johnson et al., 2017]. In recent works [Papernot and McDaniel, 2018, Dubey et al., 2019, Sitawarin and Wagner, 2019b], $k$-NN is combined with neural networks to enhance the robustness and the interpretability. Despite the importance of $k$-NN, there are only a handful of works that study its robustness [Wang et al., 2018, Yang et al., 2020, Wang et al., 2019, Sitawarin and Wagner, 2019a, 2020].

The first step towards evaluating the robustness of a classifier is to generate an adversarial example that is close to a given test point, under some definition of closeness. In the case of $k$-NN where $k > 1$, this problem is challenging as the computation involves high-dimensional polytopes that satisfy a set of geometric properties, the so-called Voronoi cells.

35th Conference on Neural Information Processing Systems (NeurIPS 2021).

In this work, we propose the GeoAdEx algorithm for finding adversarial examples for $k$-NN classifiers. GeoAdEx is the first algorithm that exploits the underpinning geometry of the $k$-NN classifier. Specifically, our approach performs a principled search on the underlying high-order Voronoi diagram by expanding the search radius around the test point until it finds an adversarial, or simply incorrect, classification. The geometric foundation of GeoAdEx allows us to locate nearby adversarial cells that the other attacks typically miss. GeoAdEx follows the footsteps of the work by Jordan et al. [2019] who exploited the geometry of neural networks for verifying the outputs. Our algorithmic approach stands in sharp contrast to previous attacks for $k$-NN [Yang et al., 2020, Wang et al., 2019] which refine the exhaustive approach by heuristic-based filtering.

Furthermore, we introduce optimizations and approximations to the main algorithm, and as a result, the experiments show that GeoAdEx discovers the smallest adversarial distance compared to all of the baselines in the vast majority of our experiments for $k \in \{3, 5, 7\}$. GeoAdEx finds up to $25\%$ smaller mean adversarial distance compared to the second best attack. We note that one inherent shortcoming of our geometric approach is the increased computation time, an aspect that can be further improved.

Finally, we present experiments that demonstrate that GeoAdEx performs significantly better when points from different classes are "mixed" together, i.e., there is no clear separation between classes. On a high level, such a setup generates a more intricate spatial tessellation with nearby cells that alternate classes. In this challenging case, GeoAdEx performs up to $5\times$ better than the baselines.

## 2 Background and Related Work

### 2.1 Background

Let $X = (x_1, \ldots, x_n)$ be a set of points from $\mathbb{R}^d$, also called *generators*, and let $Y = (y_1, \ldots, y_n)$ be the class of each point from a set of possible classes $\{1, \ldots, c\}$. Let $d(x, x')$ denote a metric between $x, x' \in \mathbb{R}^d$. Let $Z^{(k)}(X)$ be the set of all possible subsets consisting of $k$ points from $X$, i.e., $Z^{(k)}(X) = \{L_1^{(k)}, \ldots, L_i^{(k)}, \ldots, L_l^{(k)}\}$, where $L_i^{(k)} = \{x_{i1}, \ldots, x_{ik}\}$, $x_{ij} \in X$ and $l = \binom{n}{k}$. We define as *order-k Voronoi cell* associated with $L_i^{(k)}$ the set $V(L_i^{(k)})$ that includes the points of $\mathbb{R}^d$ that are closer to $L_i^{(k)}$ than any other $k$ points of $X$. More formally:

$$V(L_i^{(k)}) = \left\{ p \mid \max_x \{d(p, x) | x \in L_i^{(k)}\} \leq \min_{x'} \{d(p, x') | x' \in X \setminus L_i^{(k)}\} \right\}$$

With the term *Voronoi facet*, or simply facet, we refer to a boundary of an order-$k$ Voronoi cell. We define as bisector $B\{x_a, x_b\}$ the set of points from $\mathbb{R}^d$ that are equidistant from $x_a \in X$ and $x_b \in X$. There is a tight connection between bisectors and facets. Every Voronoi facet is part of a bisector, but not every bisector, $\binom{n}{2}$ in total, includes a facet. We call the bisectors that do include a facet *active* bisectors and those that do not *inactive* bisectors. We also note that an order-$k$ Voronoi cell has at most $k(n-k)$ facets. The collection of order-k Voronoi cells for all $L_i^{(k)}$ is called *order-k Voronoi diagram*. In this work, we assume the standard non-cocircularity assumption[1] to avoid degenerate cases. We only consider the Euclidean distance for the $k$-NN, i.e., $d(x_a, x_b) = \|x_a - x_b\|_2$.

Up to this point, we have not used the classes of the $X$. In this work, we focus on $k$-NN classifiers that classify based on the majority (class) vote of the $k$-nearest points. We define as *adversarial cell* with respect to $(x, y)$ any order-$k$ Voronoi cell that has different majority than label $y$. We denote with $A(x)$ the set of all adversarial cells with respect to $(x, y)$. With the term *adversarial facet* with respect to $(x, y)$ we refer to a facet that belongs to a cell from $A(x)$.

### 2.2 Related Work

**Adversarial Robustness of $k$-NN Classifiers.** Wang et al. [2018] study the robustness property of $k$-NN classifiers from a theoretical perspective. The authors show that a $k$-NN classifier can be as robust as the optimal Bayes classifier given a sufficiently large number of generators and $k$. Wang et al. [2018] and Yang et al. [2020] also propose potential methods for improving the robustness of $k$-NN by pruning away some of the "ambiguous" generators. Other works consider different alternatives for

---

[1] In practice this can be achieved by adding a small random noise on each generator to break potential cocircularity between them.

enhancing the robustness of neural networks by combining them with $k$-NN. [Papernot and McDaniel, 2018, Sitawarin and Wagner, 2019b, Dubey et al., 2019]

**Attacks on $k$-NN Classifiers.** Sitawarin and Wagner [2019a, 2020] propose a method for finding minimum-norm adversarial examples on $k$-NN and deep $k$-NN classifiers [Papernot and McDaniel, 2018]. Their method approximates a $k$-NN classifier with a soft differentiable function so adversarial examples can be found via a gradient-based optimization algorithm. While the approach is efficient and scalable to large $k$, it provides no guarantee for finding the nearest adversarial example and overlooks the geometry of $k$-NN.

Yang et al. [2020] and Wang et al. [2019] take a similar approach for finding the adversarial example closest to a given test $x$. Their method follows an exhaustive search by computing the distance between $x$ and all the Voronoi cells that have a different class from $x$. Both of them show that this can be done exactly when $k = 1$ but does not scale well for $k > 1$. Consequently, they introduce heuristics to improve the efficiency by choosing only a subset of the Voronoi cells. However, in the process, they lose the optimality guarantee on the resulted adversarial example. More importantly, the heuristics will miss an exponential number of cells as $k$ increases.

**Attacks Based on Geometric Insights.** The problem of verifying or finding the nearest adversarial examples in neural networks is an active research direction where solutions still do not scale to a satisfactory degree [Huang et al., 2017, Katz et al., 2017, Tjeng et al., 2017, Wong and Kolter, 2018, Weng et al., 2018, Cohen et al., 2019, Jordan et al., 2019]. Among these works, Jordan et al. [2019] proposed an alternative for a ReLU network, using a geometric approach. Specifically, the authors notice that piecewise linear networks partition the input space into numerous polytopes where points of the same polytope are classified with the same label. Armed with this observation, they propose an algorithm that iteratively searches through these polytopes that are further away from the test sample until they find a polytope with a different label.

## 3 The GeoAdEx Algorithm

**Objective.** The goal of the algorithm is to find the smallest perturbation $\delta^*$ that moves a test point $(x, y)$ to an adversarial cell. This objective can be expressed as the following optimization problem:

$$\delta^* = \underset{\delta}{\arg\min} \quad \|\delta\|_2^2 \tag{1}$$
$$\text{s.t.} \quad x + \delta \in A(x)$$

We define as $\epsilon^*$ *optimal adversarial distance* where $\epsilon^* := \|\delta^*\|_2$. The norm of any other (non-optimal) perturbation $\delta$ that misclassifies $(x, y)$, i.e., $x + \delta \in A(x)$, is simply called *adversarial distance*.

**A First Approach.** The constraint of the above formulation implies that $x + \delta$ must be a member of an adversarial cell from $A(x)$. A simple approach to solve this minimization is to build a series of optimization problems, one for each of the cells in $A(x)$, and pick the solution with the minimum adversarial distance. Unfortunately, this would require solving $O(\binom{n}{k})$ quadratic programs each of which has $k(n - k)$ constraints. While this complexity may be manageable when $k = 1$ and $n$ is small, it does not scale with $k$. On a high level, Yang et al. [2020] and Wang et al. [2019] take this approach and additionally develop heuristics to scale to cases with $k > 1$.

**The GeoAdEx Approach.** The core idea of GeoAdEx is to perform a principled *geometric exploration* around the test point $x$. GeoAdEx processes order-$k$ Voronoi cells à la breadth-first search until it discovers an adversarial cell. Algorithm 1 provides a pseudocode of the main steps of GeoAdEx, and Figure 1 illustrates some steps of GeoAdEx on a 3-NN classifier with two classes. In the following subsections, we explain these steps in detail and describe performance speedups as well as approximation steps to scale the algorithm to $k > 1$.

### 3.1 Data Structures

We maintain a priority queue $PQ$ as an auxiliary data structure that is defined for a given test point $(x, y)$. $PQ$ tracks the progress made so far towards locating the optimal adversarial distance.

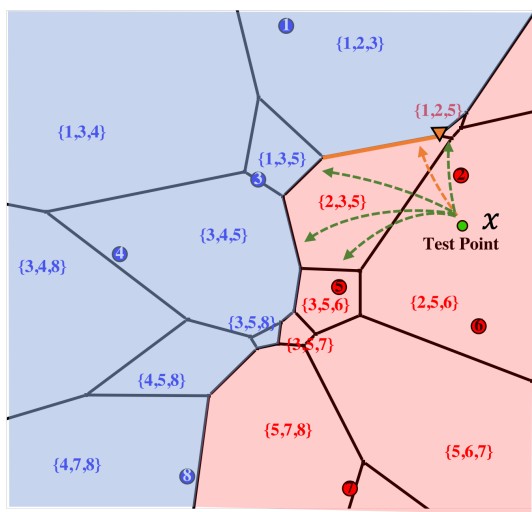

Figure 1: GeoAdEx on a Voronoi diagram for $k = 3$ in $\mathbb{R}^2$ with two classes. The color of each numbered generator point shows its class, and the color of each cell indicates its classification outcome. The illustration presents the step where GeoAdEx has processed the facets of $V(\{2, 5, 6\})$ and is transitioning to the next closest cell $V(\{2, 3, 5\})$. The arrows indicate the facets that are inserted to the priority queue $PQ$. When GeoAdEx terminates, it outputs the closest adversarial facet (orange line); this transition is indicated with the orange arrow, and the orange triangle is the optimal adversarial example for $x$.

---

**Algorithm 1:** GeoAdEx Algorithm

**Data:** Test point $(x, y)$
**Result:** Adversarial distance $\epsilon$

1  Initialize the smallest adversarial distance $\epsilon$ found so far as $\epsilon \leftarrow \infty$;
2  Initialize $PQ$ by inserting the facets of the order-$k$ Voronoi cell that $x$ falls into. Mark this cell as *visited*;
3  Call *deleteMin* from $PQ$ until the returned facet is part of an unvisited order-$k$ Voronoi cell. Initialize $\Psi$ to this unvisited Voronoi cell;
4  **while** $\Psi$ *is not an adversarial cell* **do**
5      Find the set of facets $\Phi$ that comprise unvisited cell $\Psi$ (Section 3.2);
6      Compute the distance between query point $x$ and each of the facets in $\Phi$ (Section 3.3);
7      If (i) a facet from $\Phi$ is an adversarial facet, and (ii) the distance to this adversarial facet implies a smaller norm than $\epsilon$, then update $\epsilon$;
8      Insert to $PQ$ the facets in $\Phi$ with their distance if smaller than $\epsilon$;
9      Call *deleteMin* from $PQ$ until the returned facet is part of an unvisited order-$k$ Voronoi cell. Update $\Psi$ to the new unvisited Voronoi cell;
10 **end**
11 If an adversarial facet is removed from $PQ$ via *deleteMin*, the algorithm return the optimal adversarial distance (see Lemma 1);
12 **return** $\epsilon$;

---

Specifically, $PQ$ performs the operations *insert* (resp. *deleteMin*) where the input (resp. output) is a facet of the order-$k$ Voronoi diagram of $X$. Every facet in $PQ$ is accompanied by the distance between the test point $x$ and the facet. Priority is given to the facet with the *minimum* distance to $x$. We also need to mark which Voronoi cells are processed. For that, we use a hash table (Python's native dictionary object) that has an entry for every order-$k$ Voronoi cell that the algorithm has visited. We refer to Voronoi cells that are part of the dictionary as *visited* and those that are not as *unvisited*.

## 3.2 Find Neighbors of a Voronoi Cell

Line 3 & 9 of Algorithm 1, GeoAdEx discovers an *unvisited* order-$k$ Voronoi cell $\Psi$. In Line 5, GeoAdEx needs to find the facets that comprise $\Psi$, which is equivalent to finding the Voronoi cells that are *neighboring* with cell $\Psi$. We present three options for implementing the discovery of neighbors; the choosing among them depends on the application, the size of the dataset, and the value of $k$.

**(A) Neighbors via Order-$k$ Voronoi Diagram.** The exact computation of an order-$k$ Voronoi diagram in high dimensions constitutes a computationally challenging task. The cost of this option amortizes when there is a large number of test points to be processed, and the dataset is fairly small in size and dimension. However, it is highly unlikely to scale when $k > 1$ which is the goal of our work.

**(B) Neighbors via Enumerating Bisectors.** In this approach, one has to process a quadratic number of bisectors that are potentially *active* with respect to the Voronoi cell $\Psi$ (see the definition from Section 2). Interestingly, we can filter out this set by only considering the bisectors whose generator, exactly one of the two, also generates cell $\Psi$. Consequently, the task of finding the facets of the current cell $\Psi$ is reduced to testing the *activeness* of $k(n-k)$ bisectors. Jumping ahead, this step can be accomplished simultaneously with computing distance from $x$ to each bisector so we defer its analysis for Section 3.3. For comparison, the complexity of this step is $O(poly(n, d, k))$ without any speedups, and in the worst case, which is extremely unlikely in practice, it can be called up to $N$ times where $N$ is the number of all cells ($N$ is $O(\binom{n}{k})$).

**(C) Neighbors via Order-1 Voronoi Diagram.** This approach is the middle ground between the first two. That is, it still tests whether a bisector is active, much like (B), but it uses the Voronoi diagram, much like (A), to reduce the number of bisectors to test. Given a Voronoi cell with $k$ generators, e.g., $V(\{x_1, \ldots, x_{k-1}, x_k\})$, we know that its neighboring cells must have a set of generators that differs by exactly one point, e.g., $V(\{x_1, \ldots, x_{k-1}, x_l\})$. We draw a connection between the Voronoi diagrams of order 1 and $k$ and show that the new point $x_l$ *must be neighboring with at least one of the order-1 cells* $V(\{x_1\}), V(\{x_2\}), \ldots, V(\{x_k\})$. We formalize this in Theorem 1, and its proof can be found in Appendix D.

**Theorem 1.** *Let $S = \{x_1, \ldots, x_{k-1}\} \subset X$ be a set of $k - 1$ generators. Let $x_k, x_l \in X$ be two generators such that $x_k, x_l \notin S$. If $V(S \cup \{x_k\})$ and $V(S \cup \{x_l\})$ are two neighboring order-$k$ Voronoi cells, then the order-1 Voronoi cell $V(\{x_l\})$ is neighboring with at least one of the $V(\{x_1\}), \ldots, V(\{x_{k-1}\}), V(\{x_k\})$.*

With this insight, we can *narrow down significantly* the number of bisectors considered in approach (B). More formally, let $nb(x')$ denote the set of generators of order-1 Voronoi cells that neighbor with $V\{x'\}$. Let $\Psi$ be the Voronoi cell $V(\{x_1, \ldots, x_k\})$. By Theorem 1, it is enough to consider the bisectors between $z \in \{x_1, \ldots, x_k\}$ and $z' \in \left(\bigcup_{i=1}^{k} nb(x_i)\right) - \{x_1, \ldots, x_k\}$.

Hence, the total number of bisectors we need to test is at most $k \cdot |\bigcup_{i=1}^{k} nb(x_i)|$. Assuming that $\ell := \max_i |nb(x_i)|$, the number of bisectors cannot exceed $k^2 \ell$. Hence, approach (C) is much faster than approach (B) when $k^2 \ell \ll n$. In this work, we use approach (B) for the $k = 1$ case. However, in order to scale to large datasets and $k > 1$, we propose some approximations steps that are rooted in approach (C) later in Section 3.5.2.

## 3.3 Computing Distance & Testing Activeness

Now we describe Line 6 of Algorithm 1, i.e. how to compute the distance between a point $x$ and a facet. Then, expanding on this process, we detail how to test the activeness of bisectors.

**Distance computation.** A Voronoi cell is a polytope that can also be described as an intersection of halfspaces: $\{z \mid Az \leq b\}$. A bisector can be described as a hyperplane: $\{z \mid \langle z, \hat{a} \rangle = \hat{b}\}$. The corresponding facet needs to both be part of the polytope *and* satisfy the equation of the hyperplane. Thus, the shortest distance from $x$ to a facet is given by the square root of the optimum of the following quadratic program:

$$\min_{z} \quad \|z - x\|_2^2 \tag{2}$$
$$\text{s.t.} \quad Az \leq b \text{ and } \langle z, \hat{a} \rangle = \hat{b}$$

This problem can be solved by any off-the-shelf solver, but in Appendix F, we describe a faster method also used by Wang et al. [2019].

**Testing Bisector Activeness.** Assuming that $\{z \mid Az \leq b\} \neq \emptyset$, the bisector $\langle z, \hat{a} \rangle = \hat{b}$ is active *if and only if* Eqn. (2) is feasible since their intersection has to be non-empty. In case this problem

is infeasible, the bisector is inactive. From duality theory, we know that Eqn. (2), i.e. the primal, is feasible if and only if the dual objective is unbounded since the dual problem is always feasible in this case. Checking the unboundedness can be accomplished as we are solving Eqn. (2) in its dual form so we can combine the steps of "distance computation" and "testing activeness" into a single function that either returns the distance or indicates that the bisector is inactive.

## 3.4 Optimality of GeoAdEx

Here, we formally state two lemmas that together prove the output optimality of GeoAdEx. The next lemma is straightforward as it is a direct consequence of Lemma C.2 from Jordan et al. [2019]. For completion, we prove it in Appendix D.

**Lemma 1 (Correctness of GeoAdEx).** *Provided no time limit, GeoAdEx terminates when it finds the optimal adversarial examples or equivalently, one of the solutions of Eqn.* (1).

The next lemma states that if GeoAdEx terminates early, i.e. in case we enforce a time limit for performance purposes (see Section 3.5.2), it can still guarantee a lower bound to $\epsilon^*$. Specifically, the distance to $x$ from the most recent facet deleted from $PQ$ serves as a lower bound to $\epsilon^*$.

**Lemma 2 (Lower bound guarantee).** *If GeoAdEx terminates early, the distance from test point $x$ to the last deleted facet from $PQ$ is a lower bound to the optimal adversarial distance $\epsilon^*$.*

We note that if GeoAdEx terminates early, then $\epsilon$ serves as an upper bound of $\epsilon^*$. None of the previous approaches on adversarial examples for $k$-NN provide both an upper and lower bound guarantee.

## 3.5 Towards Scaling GeoAdEx

For simplicity, in Algorithm 1, we present the main algorithmic steps that need to be performed to compute the optimal adversarial distance using the geometric structure of the problem. In this section we delve into details on how to accelerate the performance of GeoAdEx by *performance optimization* and *approximation* steps.

### 3.5.1 Optimizing Computation Time

We introduce four performance optimizations that significantly speed up GeoAdEx. We present the two most important ones here and the rest in Appendix E.

**(I) Pruning Distant Facets.** Given *any* adversarial distance, e.g., the intermediate result $\epsilon$ from Algorithm 1, we know that any facet that is more than $\epsilon$ afar from $x$ is not the facet associated with the optimal adversarial distance, i.e., $\epsilon$ acts as an upper bound that is refined during the execution. Hence, we can safely filter out these facets, and it is unnecessary to compute their distance to $x$, test activeness, or insert them to $PQ$. This technique can be used both before and during the facet distance computation (Section 3.3). A benefit of our principled geometric approach is that we can apply geometric arguments to eliminate redundant computation on Voronoi cells that are far away.

**(II) Rethinking the Initialization of $\epsilon$.** Recall that Line 1 of Algorithm 1 initializes $\epsilon$ to $\infty$. Given the upgraded role of $\epsilon$ in the previous paragraph, it is clear that a non-simplistic initialization would filter out more unnecessary computation early on and scale the overall performance. For our experiments, we run Sitawarin and Wagner [2020] to initialize $\epsilon$ since it yields a reasonable $\epsilon$ and is faster than the other attacks.

### 3.5.2 Acceleration via Approximations

As mentioned in Section 3.2, approach (C) for finding the neighbors of an order-$k$ Voronoi cell still requires the knowledge of the first-order cells. This can be obtained by either computing the entire order-1 Voronoi diagram or enumerating all possible facets of order-1 cells. Nonetheless, first-order cells in a high-dimension dataset may have a large number of neighbors, and building a Voronoi diagram can easily become a bottleneck in high-dimensions. In this case, approach (C) is no better than approach (B) of Section 3.2. To scale to large and high-dimension datasets, we introduce some approximations. Here, we propose the *approximate version* of GeoAdEx built upon the relationship between order-1 and order-$k$ neighbors from Theorem 1.

**Description.** To circumvent the expensive (in high dimensions) computation of the order-1 neighbors, we approximate approach (C) of Section 3.2. Specifically, instead of operating on the neighbors of the order-1 cell $V\{x_i\}$, we operate on a subset of $m$ points chosen from the entire set of $X$ according to a fast heuristic. In other words, $nb(x_i)$ is (roughly) approximated by a new subset of fixed size $m$ denoted as $\widetilde{nb}_m(x_i)$. For cell $V\{x_1, \ldots, x_k\}$ and a given generator $x_i$ of this cell, we select from $X$ the $m$ closest points to $x_i$ that are not in $\{x_1, \ldots, x_k\}$. Mathematically, we define $\widetilde{nb}_m(x_i)$ as:

$$\widetilde{nb}_m(x_i) = \left\{ x_j \in X \mid d(x_i, x_j) \leq d(x_i, x_{\pi(m)}) \right\} \setminus \{x_1, \ldots, x_k\}$$

where $x_{\pi(m)}$ is the $m$-th nearest neighbor of $x_i$. With the above heuristic we guarantee that for each cell, we only have to compute the distance (and test activeness) for at most $k^2 m$ facets and, thus, sidestep the computation of the first order Voronoi diagram in high dimensions. Each subsequent optimization problem has at most $k^2 m$ constraints. Finding $m$ nearest points to a single point $x_i$ is a well studied problem and can be *approximately* solved very fast [Johnson et al., 2017, Aumüller et al., 2017, Andoni et al., 2018].

**Limitations due to Approximation.** Due to the above approximation, it is possible that some of the true order-1 neighbors are not included in $\bigcup_{i=1}^{k} \widetilde{nb}_m(x_i)$. Hence, some active facets and cells may be missed completely[2]. This leads to two limitations. The first limitation is that we can no longer guarantee the optimality of GeoAdEx through Lemmas 1 and 2. In other words, we cannot conclude whether the adversarial distance returned by the approximate version of GeoAdEx is the optimal adversarial distance (or its lower bound).

The second limitation is that the approximation may affect the correctness of the distance computation. Recall that in the exact version of GeoAdEx, we have shown that it is possible to replace the bisectors in the constraint of the optimization problem in Eqn. (2) with bisectors between $\{x_1, \ldots, x_k\}$ and $\bigcup_{i=1}^{k} nb(x_i)$. But with the approximation, the new feasible set in Eqn. (2) becomes a *superset* of the one in the exact version. Consequently, the steps in Section 3.3 may falsely label an inactive bisector as active or return smaller distance than the true value.

**Addressing Limitations.** The first limitation is inherent to the deployment of heuristics to boost performance; a similar issue appears in Sitawarin and Wagner [2020] as well as the approximate version of Yang et al. [2020] and Wang et al. [2019]. It is difficult to avoid without increasing the runtime significantly. However, the second limitation can be addressed, and we do so by using the full set of bisectors when computing the distance and testing the activeness of adversarial facets like in the exact version. This incurs only a small computational cost. We do not need to do the same for non-adversarial cells since their distance and activeness do not affect the correctness of GeoAdEx.

## 4 Experiments

We compare GeoAdEx to all of the previously proposed attacks on $k$-NN classifiers, namely Sitawarin and Wagner [2020], Yang et al. [2020], and Wang et al. [2019]. The attacks are evaluated on seven datasets most of which were used in the experiments by Yang et al. [2020]. Namely, we evaluate the attacks in that following datasets: Australian, Covtype, Diabetes, Fourclass, and fmnist06 (a two-class subset, '0' and '6', of Fashion-MNIST). Additionally, we also evaluate on Gaussian and Letters datasets which have more classes and data points. More details on the datasets and the computational setup are included in Appendix A.

### 4.1 Main Findings

Table 1 compares the proposed GeoAdEx attack against the three baselines with respect to the mean perturbation norm. Interestingly, GeoAdEx *outperforms the other baselines* on all of the seven datasets for $k = 3, 5$ and on four of them for $k = 7$. In the three remaining datasets, Sitawarin and

---

[2]Depending on the dataset and $m$, the chance of this happening may not be high for two reasons: (i) For a large $k$, $\bigcup_{i=1}^{k} \widetilde{nb}_m(x_i)$ becomes a large set and is likely to cover most, if not entire, $\bigcup_{i=1}^{k} nb(x_i)$. (ii) Even if we miss some cells as neighbors of a particular order-$k$ cell, they may still be picked up by the other cells.

Table 1: Mean norm of the adversarial perturbations on 100 random test points across datasets (lower is better). We report the numbers averaged over 10 runs with random training and test splits. The error, highlighted in gray, is the 95%-confidence interval. The numbers in parentheses is the ratio of the mean perturbation norm found by GeoAdEx over that of the best baseline. The smallest mean perturbation norm among the attacks for each dataset and each $k$ is bolded.

| $k$ | Attacks | Australian | Covtype | Diabetes | Fourclass | Gaussian | Letters | fmnist06 |
|---|---|---|---|---|---|---|---|---|
| | S&W | .4242 | .1931 | .1055 | .1079 | .0442 | .1128 | .1554 |
| | | ±.0201 | ±.0148 | ±.0068 | ±.0039 | ±.0028 | ±.0049 | ±.0121 |
| | Yang et al. | .4658 | .2385 | .1392 | .1209 | .1138 | .1370 | .1773 |
| | | ±.0258 | ±.0101 | ±.0077 | ±.0055 | ±.0023 | ±.0046 | ±.0043 |
| 3 | Wang et al. | .4466 | .2134 | .1164 | .1117 | .0825 | .1218 | .1643 |
| | | ±.0205 | ±.0170 | ±.0052 | ±.0039 | ±.0030 | ±.0050 | ±.0047 |
| | **GeoAdEx** | **.3646** | **.1448** | **.0786** | **.1073** | **.0426** | **.1091** | **.1509** |
| | | ±.0250 | ±.0116 | ±.0042 | ±.0045 | ±.0015 | ±.0066 | ±.0076 |
| | | (.8595) | (.7499) | (.7450) | (.9944) | (.9638) | (.9672) | (.9710) |
| | S&W | .4748 | .2281 | .1215 | .1087 | .0463 | .1134 | .1648 |
| | | ±.0199 | ±.0121 | ±.0060 | ±.0052 | ±.0030 | ±.0064 | ±.0093 |
| | Yang et al. | .5524 | .3047 | .1824 | .1309 | .1776 | .1503 | .2087 |
| | | ±.0141 | ±.0161 | ±.0068 | ±.0043 | ±.0031 | ±.0048 | ±.0059 |
| 5 | Wang et al. | ..5110 | .2613 | .1382 | .1127 | .1195 | .1298 | .1877 |
| | | ±.0147 | ±.0110 | ±.0056 | ±.0054 | ±.0047 | ±.0055 | ±.0088 |
| | **GeoAdEx** | **.4608** | **.1856** | **.1021** | **.1066** | **.0401** | **.1130** | **.1632** |
| | | ±.0175 | ±.0218 | ±.0065 | ±.0048 | ±.0023 | ±.0021 | ±.0075 |
| | | (.9705) | (.8137) | (.8403) | (.9981) | (.8661) | (.9965) | (.9903) |
| | S&W | **.5110** | .2528 | .1259 | .1129 | .0463 | **.1127** | **.1662** |
| | | ±.0145 | ±.0211 | ±.0089 | ±.0043 | ±.0022 | ±.0049 | ±.0051 |
| | Yang et al. | .6010 | .3335 | .2005 | .1403 | .2204 | .1637 | .2328 |
| | | ±.0140 | ±.0245 | ±.0097 | ±.0053 | ±.0021 | ±.0035 | ±.0057 |
| 7 | Wang et al. | ..5410 | .3078 | .1631 | .1182 | .1570 | .1395 | .2010 |
| | | ±.0181 | ±.0145 | ±.0083 | ±.0051 | ±.0017 | ±.0034 | ±.0056 |
| | **GeoAdEx** | .5134 | **.2153** | **.1202** | **.1109** | **.0425** | .1151 | .1692 |
| | | ±.0149 | ±.0174 | ±.0049 | ±.0019 | ±.0033 | ±.0065 | ±.0055 |
| | | (1.0047) | (.8517) | (.9547) | (.9822) | (.9081) | (1.0213) | (1.0181) |

Wagner [2020] performs best and is narrowly better than GeoAdEx. GeoAdEx performs notably well on datasets Gaussian and Covtype with up to $25\%$ smaller perturbation norm than the second best attack. On average, GeoAdEx reduces the perturbation norm by $11\%, 8\%$, and $4\%$ for $k = 3, 5, 7$, respectively. Since the results of the exact attacks for $k = 1$ are not the main focus of this work, they appear in Appendix B.1.

We report that while our attack finds an adversarial distance that is closer to the optimal compared to the baselines, its main limitation is the runtime. In some cases, the runtime of GeoAdEx can be an order of magnitude larger than the second best attack. This is discussed in more detail in Section 4.3.

It is also important to note that each of the proposed attacks (including GeoAdEx) performs better when the underlying data present certain structural properties. As a result, there is no single attack, so far, that performs universally better across all datasets. In Section 4.2, we explore the property of datasets that make our attack superior.

## 4.2  Advantages of a Geometric Search

Intuitively, GeoAdEx is based on a search that expands outwards from the original test point. As a result, it performs significantly better than the baselines when there exists an adversarial cell in relatively close proximity to the given test point. In other words, GeoAdEx performs really well on datasets when the class-conditioned distributions are closer or present significant "overlap." In this case, GeoAdEx is less likely to miss small adversarial cells that are close to the test point as confirmed by the main experiment. We verify this hypothesis in the following experiment where we control the closeness of classes and study the performance of GeoAdEx compared to the baselines.

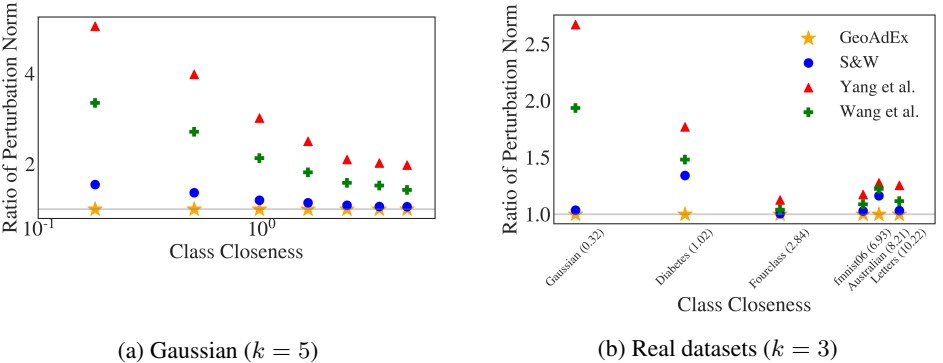

(a) Gaussian ($k = 5$)       (b) Real datasets ($k = 3$)

Figure 2: Mean perturbation norm found by the attacks as a function of the class closeness (lower is better).

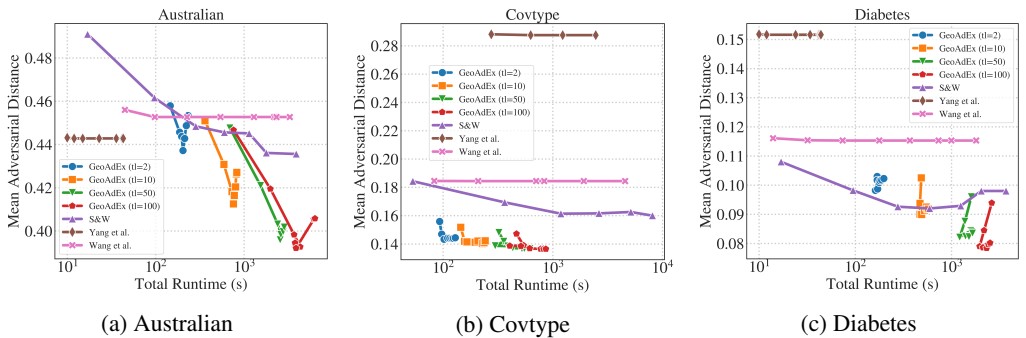

(a) Australian      (b) Covtype      (c) Diabetes

Figure 3: Mean adversarial distance vs. total runtime for GeoAdEx and all the baselines with different choices of hyperparameters on (a) Australian, (b) Covtype, and (c) Diabetes. Each point represents a single run with a unique set of hyperparameters. For a given runtime, an attack with the lowest mean adversarial distance is the best. GeoAdEx utilizes the extra runtime better and outperforms the baselines.

**Experiments with Class Closeness.** Given a dataset with $c$ class-conditioned data distributions $D_1, \ldots, D_c$, we can compute the minimum KL-divergence between a pair of distributions $D_i$ and $D_j$ : $\mathsf{KLD}(D_i||D_j)$. Now we define as *class closeness* of a given data distribution as the average of the above set of minima, i.e., $\frac{1}{c}\sum_{i=1}^{c}\min_j \mathsf{KLD}(D_i||D_j)$. The smaller the class closeness, the closer their distributions are and, consequently the larger the degree of "mixing" between Voronoi cells of different classes. For each dataset, we record the associated class closeness and the ratio of the mean perturbation norm of each baseline over that of GeoAdEx.

First, we confirm this hypothesis on synthetic datasets where each of the two classes is generated by a Gaussian distributions, i.e., $D_1, D_2$, in $\mathbb{R}^{20}$. Note that for Gaussian, KL-divergence can be analytically computed as the closed form exists. In this case, Figure 2a shows that GeoAdEx *significantly outperforms* all the baselines by a larger margin when the class closeness is small. That is, for datasets that are challenging to classify, GeoAdEx outperforms the competition.

Furthermore, we revisit the real datasets from Table 1 and re-interpret the results under the lens of their class closeness. Since the true data distributions of these datasets are unknown, we approximate the KL-divergence via data samples using the estimator in Equation (5) from Wang et al. [2009]. Indeed, Figure 2b shows that the datasets on which GeoAdEx clearly outperforms Yang et al. [2020] and Wang et al. [2019], are the ones with small class closeness (e.g., Diabetes, Gaussian).

### 4.3 Runtime Comparisons

Runtimes of the experiments in Table 1 are reported in Table 4. Through the choice of hyperparameters, we attempt to control the runtime of each algorithm to roughly be within the same order of magnitude, but this has proven difficult given that we do not wish to fine-tune the hyperparameters specifically per dataset. To better compare the algorithms under a similar set of runtimes, we repeat the experiments with different sets of hyperparameters chosen at certain intervals and plot the mean adversarial distance vs. the total runtime for each experiment in Figure 3.

For each algorithm, we started with its default hyperparameters and adjusted them (either linearly or exponentially) in the direction that should find smaller adversarial perturbations with a longer runtime. Each point on the curves represents an experiment with one set of hyperparameters. One way to read the plot is to fix a particular runtime and compare the mean adversarial distance from each line. This experiment is expensive so we only run it for the first three datasets from Table 1.

For Australian and Diabetes, in a regime with very short runtime ($\sim$100s total or $\sim$1s per sample), there is at least one baseline that is both faster and finds smaller adversarial perturbation than GeoAdEx. However, with longer runtimes, our GeoAdEx outperforms all the baselines by a large margin (10%, 15% and 20% improvement over the second best for Australian, Covtype, and Diabetes, respectively). In most settings, the baselines do not benefit much, if at all, from the increased runtime. Conversely, GeoAdEx always finds smaller adversarial distances given a longer time limit. More details and discussions of this experiment are in Appendix B.2.

**Effect of the initialization attack.** The initialization of $\epsilon$ with Sitawarin and Wagner [2020] attack, as described in the performance optimization (II), also improves the total runtime of GeoAdEx. Depending on the dataset, the runtime reduction range from 2% to 75%. In the interest of space, ablation study and analysis of the initialization attack is included in Appendix B.3.

## 5 Discussion & Open Problems

GeoAdEx has been shown to be successful in discovering adversarial examples for $k$-NN classifiers with $k \geq 1$. Compared to the baselines, it finds adversarial examples with a considerably smaller perturbation norm on most of the commonly tested datasets.

The main limitation of GeoAdEx is its runtime, particularly in the short-runtime regime. Sitawarin and Wagner [2020] is always the fastest, generally followed by Wang et al. [2019], Yang et al. [2020], and then GeoAdEx. However, in the long-runtime regime, our GeoAdEx outperforms the baselines by a large margin given the same runtime. GeoAdEx utilizes a longer runtime by expanding the search radius while the baselines search more cells without a particular order or prioritization. As a result, the adversarial distance found by Sitawarin and Wagner [2020], Yang et al. [2020], and Wang et al. [2019] typically does not improve much further with more computation.

We note that compared to the geometric approach by Jordan et al. [2019], our experiments were run on a significantly larger scale. Specifically, Jordan et al. [2019] was tested on neural networks with only up to 70 ReLUs, which are much smaller than typical networks used in practice. This is equivalent to Voronoi cells with only 70 neighbors/facets in our setting. GeoAdEx was tested on datasets with over ten thousand generator points and $k$ up to 7, which results in a substantially larger number of polytopes to search through.

GeoAdEx has shown convincing results with our new geometric take on the problem, but there is room for improvement in terms of efficiency. Additional speedups can be introduced via parallelization, GPU utilization, and a faster optimization technique. A more sophisticated heuristic for determining the order-1 neighbors can also be used. Furthermore, geometric properties of a high-order Voronoi diagram may be better exploited to save unnecessary computation on non-adversarial cells. For instance, multiple neighboring non-adversarial cells may be combined and approximated as a single large cell if we can guarantee that there is no adversarial cell inside this new polytope. This would remove a large number of distance computations surrounding the given test point, which is the main bottleneck of GeoAdEx. Additionally, GeoAdEx can also be extended to other space-partitioning classifiers such as decision trees and random forests.

## 6 Conclusion

We propose GeoAdEx, an algorithm based on geometric insights for finding adversarial examples on $k$-NN classifiers. We leverage the structural properties of higher-order Voronoi diagrams to propose efficient approximations and speedup the final algorithm. While GeoAdEx typically requires a higher computational cost, it significantly outperforms the baselines in most of the datasets. Finally, for the case where there is no clear separation between the classes (a typical characteristic of real data) GeoAdEx significantly outperforms the competition.

**Acknowledgement**

The first author of this paper was supported by the Hewlett Foundation through the Center for Long-Term Cybersecurity (CLTC) and by generous gifts from Open Philanthropy and Google Cloud Research Credits program with the award GCP19980904.

The second and third authors were supported by the Center for Long-Term Cybersecurity (CLTC), the Berkeley Deep Drive project, NSF grant TWC-1518899, and Open Philanthropy.

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
