# Appendix

Appendix A contains details on the experiments conducted throughout this paper. In Appendix B, we include additional results from the experiments on GeoAdEx, e.g., attacks on $k$-NN with $k = 1$, runtime comparisons, an ablation study, and different hyperparameter choices. In Appendix C, more details on the class closeness metric are provided. In Appendix D, we provide the proofs of the theory and lemmas stated in the paper. Appendix E explains all the performance optimization used in GeoAdEx, and lastly, in Appendix F, we describe the optimization algorithm, greedy coordinate ascent, used for the distance computation.

## A   Details of the Experiments

**Datasets.**   Details regarding the datasets used in the experiments are included in Table 2. It also includes the accuracy of $k$-NN classifiers at $k = 1, 3, 5, 7$. Australian, Covtype, Diabetes, Fourclass, and fmnist06 are taken directly from Yang et al. [2020]'s implementation. The dataset fmnist06 is a two-class subset of Fashion-MNIST with a dimension reduction to 25 via PCA. The Letters dataset, together with the others, is taken from LIBSVM [Chang and Lin, 2011][3]. Gaussian is a dataset we create by sampling from two isotropic Gaussian distributions of 20 dimension and variance of 1. The distance between the means of the two distributions is 1 by default and is varied only in Section 4.2 to get different values of class closeness.

**Environment and implementation.**   All of the attacks are run on an Ubuntu (16.04) cluster with 128 AMD EPYC 7551 CPU cores (2.5GHz each) and 252 GB of memory. No GPU is used in any of the experiments. Using GPU could further speed up the attacks, but the official implementation of the baselines is not compatible with GPUs so we stick to CPUs to present a fair comparison. Yang et al. [2020] uses explicit parallelization and Cython C-extensions, whereas the rest of the attacks use pure Python code without explicit parallelization. The code for the baselines are taken directly from their respective public repository.[4] Yang et al. [2020] uses Gurobi as the solver.

**Hyperparameters.**   We evaluate all the baselines using their publicly available code and default hyperparameters. For fairness, we also attempt to tune the hyperparameters for each baseline to keep the total runtime comparable across attacks (see Appendix 4.3). In general, the baselines are fairly insensitive to changes in their hyperparameters. For example, increasing the number of Voronoi cells searched by Yang et al. [2020] and Wang et al. [2019] almost never reduces the mean perturbation norm beyond one obtained with the default value. For GeoAdEx, we choose to compute distance to cell and set $m$ to 20 and applied a time limit of 100 seconds per test point.

Table 2: Details of the datasets used in the experiments.

| Datasets | # points | # features | # classes | $k = 1$ acc | $k = 3$ acc | $k = 5$ acc | $k = 7$ acc |
|---|---|---|---|---|---|---|---|
| Australian | 490 | 14 | 2 | 0.805 | 0.805 | 0.830 | 0.845 |
| Covtype | 2000 | 54 | 7 | 0.755 | 0.715 | 0.730 | 0.705 |
| Diabetes | 568 | 8 | 2 | 0.695 | 0.755 | 0.695 | 0.685 |
| Fourclass | 662 | 2 | 2 | 1.000 | 0.995 | 1.000 | 1.000 |
| Gaussian | 10000 | 20 | 2 | 0.550 | 0.660 | 0.640 | 0.635 |
| Letters | 15000 | 16 | 26 | 0.925 | 0.940 | 0.940 | 0.930 |
| fmnist06 | 12000 | 25 | 2 | 0.800 | 0.795 | 0.810 | 0.810 |

## B   Additional Results

### B.1   Exact Attacks for $k = 1$

For completeness, we also compare the exact version of the attacks on 1-NN where the results are presented in Table 3. Note that Sitawarin and Wagner [2020] is excluded since it does not have an

---

[3]https://www.csie.ntu.edu.tw/~cjlin/libsvmtools/datasets/multiclass.html
[4]Sitawarin and Wagner [2020]: https://github.com/chawins/knn-defense, Yang et al. [2020]: https://github.com/yangarbiter/adversarial-nonparametrics, Wang et al. [2019]: https://github.com/wangwllu/knn_robustness

Table 3: Runtime for the exact version of the attacks on all of the datasets with $k = 1$. Sitawarin and Wagner [2020] is not included because it does not offer an exact solution and provide no guarantee on the adversarial distance.

| Attacks | Australian | Covtype | Diabetes | Fourclass | Gaussian | Letters | fmnist06 |
|---|---|---|---|---|---|---|---|
| Yang et al. [2020] | 72 | 7186 | 66 | 59 | 9109 | 34997 | 18461 |
| Wang et al. [2019] | 2 | 10 | 2 | 25 | 159 | 78 | 333 |
| **GeoAdEx** | 17 | 38 | 15 | 39 | 476 | 7450 | 19307 |

Table 4: Runtime (in seconds) for each of the attacks on all of the seven datasets. The mean adversarial distance corresponding to these runtimes are shown in Table 1. The numbers in the gray rows are 95%-confidence interval from 10 runs with random splits between training and testing samples.

| $k$ | Attacks | Australian | Covtype | Diabetes | Fourclass | Gaussian | Letters | fmnist06 |
|---|---|---|---|---|---|---|---|---|
| 3 | S&W [2020] | 654 | 1225 | 465 | 336 | 811 | 3372 | 972 |
| | | ±1 | ±250 | ±121 | ±5 | ±14 | ±13 | ±179 |
| | Yang et al. [2020] | 8 | 925 | 9 | 7 | 599 | 1555 | 2151 |
| | | ±1 | ±97 | ±1 | ±1 | ±6 | ±38 | ±62 |
| | Wang et al. [2019] | 361 | 363 | 128 | 129 | 226 | 259 | 272 |
| | | ±21 | ±20 | ±9 | ±5 | ±19 | ±19 | ±21 |
| | **GeoAdEx** | 3351 | 727 | 1987 | 1424 | 2525 | 4030 | 6427 |
| | | ±447 | ±100 | ±152 | ±106 | ±146 | ±354 | ±338 |
| 5 | S&W [2020] | 654 | 1225 | 465 | 336 | 811 | 3372 | 972 |
| | | ±1 | ±250 | ±121 | ±5 | ±14 | ±13 | ±179 |
| | Yang et al. [2020] | 8 | 925 | 9 | 7 | 599 | 1555 | 2151 |
| | | ±1 | ±97 | ±1 | ±1 | ±6 | ±38 | ±62 |
| | Wang et al. [2019] | 361 | 363 | 128 | 129 | 226 | 259 | 272 |
| | | ±21 | ±20 | ±9 | ±5 | ±19 | ±19 | ±21 |
| | **GeoAdEx** | 3351 | 727 | 1987 | 1424 | 2525 | 4030 | 6427 |
| | | ±447 | ±100 | ±152 | ±106 | ±146 | ±354 | ±338 |
| 7 | S&W [2020] | 654 | 1225 | 465 | 336 | 811 | 3372 | 972 |
| | | ±1 | ±250 | ±121 | ±5 | ±14 | ±13 | ±179 |
| | Yang et al. [2020] | 8 | 925 | 9 | 7 | 599 | 1555 | 2151 |
| | | ±1 | ±97 | ±1 | ±1 | ±6 | ±38 | ±62 |
| | Wang et al. [2019] | 361 | 363 | 128 | 129 | 226 | 259 | 272 |
| | | ±21 | ±20 | ±9 | ±5 | ±19 | ±19 | ±21 |
| | **GeoAdEx** | 3351 | 727 | 1987 | 1424 | 2525 | 4030 | 6427 |
| | | ±447 | ±100 | ±152 | ±106 | ±146 | ±354 | ±338 |

exact version. Wang et al. [2019] is generally the fastest, and GeoAdEx is faster than Yang et al. [2020], which does not seem to scale well with the number of generators and dimension. This is a direct effect of the solvers of the quadratic programs. Greedy coordinate ascent, used by Wang et al. [2019] and our attack, is much more efficient than a general-purpose commercial solver.

## B.2 Runtime Comparisons and Attack Hyperparameters

Table 4 includes the runtime of all the attacks for $k = 3, 5, 7$. As we have mentioned, GeoAdEx with $m = 20$ and the time limit of 100 seconds takes longer to run compared to the other attacks for most cases. Sitawarin and Wagner [2020] is the fastest, followed by Wang et al. [2019] and Yang et al. [2020]. For the other cases, Yang et al. [2020] has the longest runtime. The reported runtime of GeoAdEx also includes the time used to initialize the adversarial distance $\epsilon$ found by Sitawarin and Wagner [2020].

As mentioned in Section 4.3, we conduct more thorough sets of experiments on the first three datasets (Australian, Covtype, and Diabetes) to fairly compare the attacks under a similar runtime. To this end, we plot the runtime vs. mean adversarial distance curves for GeoAdEx and all the baselines by varying their hyperparameters in Figure 3.

### B.2.1 Runtime Experiment Setup

For GeoAdEx, we vary $m$ with four different values of time limit per sample which result in four different curves. For Yang et al. [2020], we progressively doubled the number of regions searched which is the only adjustable hyperparameter. For Wang et al. [2019], we progressively doubled the number of trials (both min and max) and the number of neighbors to consider (until it exceeds the number of all generators). There are four hyperparameters for Sitawarin and Wagner [2020]: `binary_search_steps`, `max_iterations`, `thres_steps`, and `check_adv_steps`. We progressively increased the first two and decreased the last two linearly. For GeoAdEx, we tested a more detailed breakdown by varying both (from $5$ to $120$) and time limit.

Note that the runtime can be slightly different from what reported in Table 4 since we use a different machine. To make this figure, we run all the experiments (all attacks, datasets, and choices of hyperparameters) on a server with 40 cores of Intel(R) Xeon(R) Gold 6230 CPU @ 2.10GHz.

### B.2.2 Discussion on Figure 3

The major trends have already been covered in Section 4.3. Here, we discuss other observations and minor trends.

**Increasing $m$ in GeoAdEx can either increase or decrease the mean adversarial distance.** Each curve for GeoAdEx is generated by increasing $m$ but fixing the time limit. Increasing $m$ reduces the chance of missing the adversarial facets (hence the downward trend in the adversarial distance), but it also increases the computation time for each cell which means that there are fewer cells it can search given a fixed time limit (hence the upward trend). This implies that there is an optimal value of $m$ for a given time limit.

**Increasing $m$ in GeoAdEx can either increase or decrease the total runtime.** This outcome is seemingly perplexing than the previous one. We explain it for different values of $m$, namely the small-$m$ and the large-$m$ regions.

*Small-$m$ region.* When a smaller $m$ is used with GeoAdEx, fewer first-order neighbors are considered, and thus, the search has a higher chance of missing facets and nearby adversarial cells completely. As a result, it has to expand the search radius which, in turn, would discover adversarial examples that are further away and use a longer runtime. Conversely, if we increase $m$, we may find these previously missed cells and terminate earlier, resulting in both lower adversarial distance and runtime. We call the first scenario from the above exposition, the "small-$m$ region."

*Large-$m$ region.* On the other hand, when $m$ is sufficiently large and no adversarial cell is missed, increasing $m$ could have a reversed effect. In particular, when $m$ increases, more first-order neighbors have to be considered, and hence more *nearby* cells will have to be searched. For each of the test samples, this could lead to (i) an increased runtime and/or (ii) the previously found adversarial cells that are further away may now be missed instead. When (ii) happens, GeoAdEx will timeout and just return the initialized upper bound. Therefore, in the "large-$m$ region," both the adversarial distance and the total runtime may increase with $m$.

We verify this hypothesis by inspecting the number of samples that are timed out by GeoAdEx. If our hypothesis holds, when we test different values of $m$, we expect to see a decreasing trend on the number of timeouts in the small-$m$ region and an increasing trend in the large-$m$ region. Specifically, when varying $m \in \{5, 10, 20, 40, 60, 80, 100, 120\}$ on the Diabetes dataset, we observe the following number of timeouts over 100 samples: 41, 21, 14, 16, 10, 15, 16, and 17. The first three experiments ($m = 5, 10, 20$) which correspond to the small-$m$ region show a decreasing number of timeouts from 41 to 14. The last three experiments ($m = 80, 100, 120$) correspond to the large-$m$ region where both runtime and distance increase with $m$. The same phenomenon also happens on Covtype given the same hyperparameters and the range of values of $m$. In this case, the numbers of timeouts are 24, 9, 3, 3, 3, 4, 4, and 5, respectively.

Note that whether a value of $m$ is considered "small" or "large" varies by datasets and the time limit. Increasing the time limit reduces the number of timeouts and hence delays the large-$m$ region (i.e., occurs at a larger $m$). Additionally, this observation leads to two practical suggestions: (1) regardless of $m$, increasing the time limit is always beneficial in terms of the adversarial distance, but (2) for a fixed time limit, there is an optimal value of $m$.

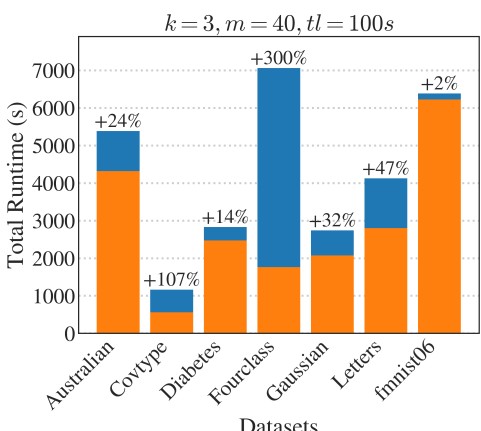

Figure 4: Improvement in the total runtime of GeoAdEx with (orange) and without (blue) Sitawarin and Wagner [2020] initialization.

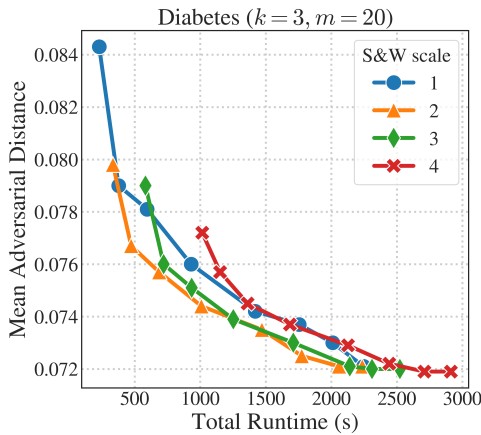

Figure 5: Mean adversarial distance vs. total runtimes on GeoAdEx using S&W initialization with four different hyperparameter scaling (corresponding to the four curves). Each point on the curve represents a unique choice of the time limit per sample in GeoAdEx (from 5 to 200 seconds).

**Curves for Yang et al. [2020] are shorter than the others.** For Australian and Diabetes, the lines associated with Yang et al. [2020] are shorter than the rest because we cannot increase the total runtime by adjusting the hyperparameter any further. This is a fundamental flaw of the heuristic used by Yang et al. [2020] which only searches the cells that contain any generator from a wrong class. So the total number of cells searched is upper bound by the number of generators from a wrong class which is very limited compared to the total number of cells.

## B.3 Ablation Study

### B.3.1 Importance of S&W Initialization

We want to compare GeoAdEx with and without the S&W initialization. Figure 4 compares the total runtime of GeoAdEx on all of the datasets with $k = 3$ and shows that the S&W initialization speeds up the attack in all cases. While effects of the initialization are generally minor, we found two cases, namely Covtype and Fourclass, where the initialization leads to a large improvement in the runtime. Without the initialization, the runtime increases by $107\%$ and $300\%$ in these two datasets respectively. We give our explanation for these two datasets below.

For Covtype, GeoAdEx's runtimes are 561s with the initialization step and 1163s without. Without an initial upper bound on $\epsilon$, there is a small chance that GeoAdEx misses a nearby adversarial cell and keeps running until the time limit is met. While only several test samples are affected, it ends up raising the total runtime by a relatively large margin ($100\%$ from 500s) because the time limit is set to 100s per sample. The initialization has the same kind of effect on Fourclass as on Covtype, but it affects a much larger number of samples. The number of timeouts goes from 4 to 43 as the initialization is removed, explaining the significant increase in the total runtime. One hypothesis is that this phenomenon takes place because S&W attack is particularly effective on Fourclass (note its competitive performance compared to the other attacks), and thus removing it as an initialization results in an equally large degradation on the performance of GeoAdEx.

### B.3.2 Effects of S&W Initialization's Hyperparameters

To test the effect of the initialization's hyperparameters on GeoAdEx, we ran a simple experiment on the Diabetes dataset by varying the runtime of S&W initialization and the time limit of GeoAdEx. The trade-off curves between total runtime and the adversarial distance are shown in Figure 5. Each curve uses a different scaling factor of the S&W attack as described in our previous experiment to vary its runtime. Each dot in the curve is generated by varying the time limit of GeoAdEx from 5 to 200 seconds. For a wide range of total runtimes, S&W initialization with a scaling factor between 2 and 3 (or anywhere between $4\%$ and $35\%$ of the total runtime) seems like an optimal spot. This

suggests that performance of GeoAdEx is fairly insensitive to the hyperparameters of the S&W initialization. Based on this observation alone, one may aim to run the initialization for $\sim 10-20\%$ of the total runtime as the simplest baseline and avoid any additional overhead.

We simply choose S&W attack because it finds an adversarial example quickly (even though the upper bound is relatively loose) with little impact on the total runtime. The choice is somewhat arbitrary and can be replaced with the attacks by Yang et al. [2020] or Wang et al. [2019]. Deciding how long to run S&W initialization is a good question from a practical standpoint. It also depends on many factors including the dataset, the desired time limit, and hyperparameters of GeoAdEx. So it should be considered on a case-by-case basis, perhaps by a hyperparameter optimization on a small subset of the data. A cheaper strategy that works across all datasets is to terminate the initialization attack (per-sample) when the adversarial distance starts to plateau (e.g, less than $A\%$ improvement in the last $B$ iterations).

## C  Class Closeness

Intuitively, the *class closeness* measures distance between one distribution to another that has a different class and is closest to it. We only consider the closest class because when generating an adversarial example, one only has to perturb the test point towards the nearest distribution with a different class and the other classes are almost irrelevant. More formally, we can write the class closeness as

$$\text{class closeness} := \frac{1}{c} \sum_{i=1}^{c} \min_{j \in \{1,...,c\}} \mathsf{KLD}(D_i || D_j)$$

where $c$ is the number of classes, and $D_i$ is the distribution conditioned on class $i$.

We first experiment with the Gaussian dataset because its KL-divergence has a analytical form. Specifically, the KL-divergence between two multivariate Gaussian distributions in $\mathbb{R}^d$, $D_1 = \mathcal{N}(\mu_1, \Sigma_1)$ and $D_2 = \mathcal{N}(\mu_2, \Sigma_2)$, is given by

$$\mathsf{KLD}(D_1 || D_2) = \frac{1}{2} \left[ \log \frac{|\Sigma_2|}{|\Sigma_1|} - d + \text{tr}(\Sigma_2^{-1} \Sigma_1) + (\mu_2 - \mu_1)^\top \Sigma_2^{-1} (\mu_2 - \mu_1) \right]$$

In particular, we use isotropic Gaussian distributions so the means and the covariance matrices can be simplified even further.

$$\mu_1 = \begin{bmatrix} \alpha \\ 0 \\ \vdots \\ 0 \end{bmatrix}, \mu_2 = \begin{bmatrix} -\alpha \\ 0 \\ \vdots \\ 0 \end{bmatrix}, \Sigma_1 = \Sigma_2 = I_d$$

Note that we pick $d = 20$, and without loss of generality, we can simply assign different values of $\alpha$ and $-\alpha$ to the first coordinate of $\mu_1$ and $\mu_2$ to vary the distance between the two means. We pick $\alpha$ among $\{0.3, 0.5, 0.7, 0.9, 1.1, 1.3, 1.5\}$. This specific case yields a very simple form of KL-divergence:

$$\mathsf{KLD}(D_1 || D_2) = 2\alpha^2$$

For the second part, since the distributions of the other datasets are unknown, we use a non-parametric method from Wang et al. [2009] to approximate the KL-divergence. This method only requires samples from the distributions and is coincidentally based on $k$-NN. We pick $k = 5$ for this approximation method which has nothing to do with the value of $k$ in $k$-NN classifiers we experiment with.

## D  Proofs

### D.1  Theorem 1

Now we restate Theorem 1 and then the proof.

**Theorem 1.** *Let $S = \{x_1, \ldots, x_{k-1}\} \subset X$ be a set of $k-1$ generators. Let $x_k, x_l \in X$ be two generators such that $x_k, x_l \notin S$. If $V(S \cup \{x_k\})$ and $V(S \cup \{x_l\})$ are two neighboring order-$k$ Voronoi cells, then the order-1 Voronoi cell $V(\{x_l\})$ is neighboring with at least one of the $V(\{x_1\}), \ldots, V(\{x_{k-1}\}), V(\{x_k\})$.*

*Proof.* Let $V(x'|G)$ denote the order-1 Voronoi cell for $x'$ on the set of generators $G$. From property OK1 in Section 3.2.1 of Okabe et al. [1992] we know that the order-$k$ Voronoi cell $V(S \cup \{x_i\})$ can be expressed as:

$$V(S \cup \{x_i\}) = \left( \bigcap_{l=1}^{k-1} V(x_l | (X \setminus (S \cup \{x_i\})) \cup \{x_l\}) \right) \cap V(x_i | X \setminus S)$$

From the fact that $V(S \cup \{x_i\})$ is a order-$k$ Voronoi cell, we know that $V(S \cup \{x_i\})$ is nonempty. Let us assume for the sake of contradiction that $V(\{x_i\})$ is not neighboring with any of the $V(\{x_1\}), \ldots, V(\{x_{k-1}\})$. Then we know that:

$$V(\{x_i\}) = V(x_i | X \setminus S)$$

This is because the removal of $S$ from the set of generators of the Voronoi diagram did not affect $V(\{x_i\})$ since $V(\{x_i\})$ is not neighboring with any of the $V(\{x_1\}), \ldots, V(\{x_{k-1}\})$. Additionally, the removal of $x_i$ from the set of generators in the term $V(x_l | x_l \cup (X \setminus S \cup \{x_i\}))$ is not affecting the corresponding cell, again, because $V(\{x_i\})$ is not neighboring with $V(\{x_1\}), \ldots, V(\{x_{k-1}\})$. This implies that:
$$V(x_l | (X \setminus (S \cup \{x_i\})) \cup \{x_l\}) = V(x_l | (X \setminus S) \cup \{x_l\})$$

Using the above observations we can rewrite the first relation as:

$$V(S \cup \{x_i\}) = \left( \bigcap_{l=1}^{k-1} V(x_l | (X \setminus S) \cup \{x_l\}) \right) \cap V(\{x_i\})$$

For the last part of the proof we will show that the above intersection is empty which contradicts the fact that $V(S \cup \{x_i\})$ is a nonempty Voronoi cell. Notice that the changes in the set of generators that take place in the term $V(x_l | (X \setminus S) \cup \{x_l\})$ for $l = [1, k-1]$ do not affect the Voronoi cell of $x_i$. This means that even after the changes in the set of generators, the cell of $x_i$ is a superset of $V(\{x_i\})$, or to put it differently, the polytope $V(x_l | (X \setminus S) \cup \{x_l\})$ never enters the area of $V(\{x_i\})$. As a result, none of these terms intersects with $V(\{x_i\})$. But this contradict the fact that $V(S \cup \{x_i\})$ is nonempty.

$\square$

### D.2 Lemma 1: Correctness of GeoAdEx

**Lemma 1 (Correctness of GeoAdEx).** *Provided no time limit, GeoAdEx terminates when it finds the optimal adversarial examples or equivalently, one of the solutions of Eqn. (1).*

Lemma 1 can be obtained directly from a similar theorem in Jordan et al. [2019]. We first restate this theorem and the definition of *polyhedral complex*. Then, we provide a short proof.

**Theorem 2 (Correctness of GeoCert).** *(Restate from Theorem C.2 in Jordan et al. [2019]) For a fixed polyhedral complex $\mathscr{P}$, a fixed input point $x_0$ and a potential function $\phi$ that is ray-monotonic, GeoCert returns a boundary facet with minimal potential $\Phi$.*

**Definition 1 (Polyhedral Complex).** *(Restate from Definition 2 in Jordan et al. [2019]) A nonconvex polytope, described as the union of elements of the set $\mathscr{P} = \mathcal{P}_1, \ldots, \mathcal{P}_k$ forms a polyhedral complex if, for every $\mathcal{P}_i, \mathcal{P}_j \in \mathscr{P}$ with nonempty intersection, $\mathcal{P}_i \cap \mathcal{P}_j$ is a face of both $\mathcal{P}_i$ and $\mathcal{P}_j$.*

*Proof.* In order to apply Theorem 2 to GeoAdEx, we need to show two things: (i) the test point lies in a polyhedral complex, and (ii) Euclidean distance is a ray-monotonic potential function. First, it is trivial to see that the set of non-adversarial Voronoi cells connected to and including the cell the

test input $x$ falls into forms a polyhedral complex. Since Voronoi cells are polytopes and any pair of them intersect at most at the shared facet, any set of Voronoi cells forms a polyhedral complex. This (informally) proves part (i).

For part (ii), we refer the readers to Corollary C.3 of Jordan et al. [2019] which shows that Euclidean distance is a ray-monotonic potential function. With these two conditions in mind, GeoAdEx behaves in the same way as GeoCert algorithmically in their respective settings, and so Theorem 2 directly applies to GeoAdEx as well. □

### D.3 Lemma 2: Lower Bound of the Optimal Adversarial Distance

**Lemma 2 (Lower bound guarantee).** *If GeoAdEx terminates early, the distance from test point $x$ to the last deleted facet from $PQ$ is a lower bound to the optimal adversarial distance $\epsilon^*$.*

**Theorem 3.** *(Restate from Lemma C.1 in Jordan et al. [2019]) For any polyhedral complex $\mathscr{P}$ point $x_0$, and ray-monotonic potential $\phi$, let $\mathcal{F}_i$ be the facet popped at the $i$-th iteration of GeoCert. Then for all $i < j, \Phi(\mathcal{F}_i) < \Phi(\mathcal{F}_j)$.*

*Proof.* From Lemma 1, the first adversarial facet deleted from $PQ$ is the nearest one to $x$, and if that happens, GeoAdEx terminates. It is implied by Theorem 3 that the facets are always deleted from $PQ$ in an ascending order of their distance to $x$. Combining these two facts, we can conclude that the distance of any facets deleted before the adversarial one is always smaller than $\epsilon^*$. □

## E GeoAdEx Performance Optimization

We introduce a total of four performance optimizations to speed up the computation of GeoAdEx. We have explained the first two in Section 3.5 and will describe all of them here in more details.

### E.1 Pruning Distant Facets

This was described in the main text. So here, we only provide examples of where the pruning can occur to remove unnecessary facet computation. First, before computing the distance between $x$ and a facet, as proposed in Section 3.3, we can first use $\epsilon$ to filter unnecessary computation. Specifically, we can compute the orthogonal projection of $x$ onto the bisector implied by the facet. If the distance to the bisector, which is a lower bound on the distance to the facet, is larger than $\epsilon$, then this facet can be safely discarded.

Even if we proceed with the distance computation, we can still use $\epsilon$ to terminate the optimization in Eqn. (2) early. Specifically, if the dual objective of Eqn. (2) surpasses $\epsilon^2$, we can terminate the solver and discard this facet since, from strong duality, the dual objective is a lower bound of the primal objective, i.e. the (squared) distance between $x$ and this facet.

### E.2 Rethinking the Initialization of $\epsilon$

Recall that Line 1 of Algorithm 1 initializes $\epsilon$ to $\infty$. Given the upgraded role of $\epsilon$ in the previous paragraph, it is clear that a non-simplistic initialization would filter out more unnecessary computation early on and, thus, scale the overall performance. A natural choice is to pick one of the baseline attacks for this purpose. The closer this adversarial distance is to the optimal one, the more computation we are likely to save by the first performance optimization, (I) Pruning Distant Facets.

However, there is a trade-off between the tightness of the estimates and its computation time. Using an expensive attack to initialize $\epsilon$ can be a huge overhead that increases the total runtime rather than reduces it. For our experiments, we run Sitawarin and Wagner [2020] to initialize $\epsilon$ since it yields a reasonable and is significantly faster than Yang et al. [2020] and Wang et al. [2019].

### E.3 Exploiting the Sparsity of Solutions

Solving a typical quadratic program has a complexity of $O(poly(n, d, k))$, but fortunately, this problem can be solved very efficiently in its dual form. Wang et al. [2019] show that solving Eqn. (2) via greedy coordinate ascent (GCA) is much faster than using a standard off-the-shelf solver as

it is able to exploit the sparsity in the solution. More details about this speedup can be found in Appendix F.

### E.4 Setting a Time Limit

To ensure that GeoAdEx terminates in a reasonable time even when no adversarial facet has been deleted from $PQ$, we set a time limit as a termination criterion. In this case, lower and upper bounds of $\epsilon^*$ are returned instead. Note that the approximate version of Yang et al. [2020] and Wang et al. [2019] also terminates early by setting the maximum number of adversarial cells to search through instead of a limit on the runtime.

## F Distance Computation with Greedy Coordinate Ascent

We first restate the distance computation from Eqn. (2):

$$\min_{z} \quad \|z - x\|_2^2$$
$$\text{s.t.} \quad Az \leq b$$

Note that without loss of generality, the equality constraint $\langle z, \hat{a} \rangle = \hat{b}$ can be subsumed by the inequality. Now we provide the dual form of Eqn. (2):

$$\max_{\lambda} \quad g(\lambda) := -\frac{1}{2}\lambda^\top A A^\top \lambda + \lambda^\top (Ax - b)$$
$$\text{s.t.} \quad \lambda \geq 0$$

In the case that the primal and the dual problems are feasible, we know that strong duality holds because the objective is convex quadratic, and the constraints are affine [Boyd and Vandenberghe, 2004]. Thus, by setting the derivative of the Lagrangian to zero, we have that $z^* = x + A^\top \lambda^*$.

According to the complementary slackness from the KKT conditions, we know that $\lambda_i^* \neq 0$ if and only if $\langle a_i, z^* \rangle = b_i$, which geometrically corresponds to $z^*$ lying on the $i$-th bisector associated with $a_i$ and $b_i$. Intuitively, it is unlikely that $z^*$ lies on an intersection of many bisectors. Hence, there should be very few indices $i$ such that $\lambda_i^* \neq 0$. This is the condition that makes solving the dual problem with greedy coordinate ascent very fast [Wang et al., 2019].

Greedy coordinate ascent (or descent) optimizes the variable only one coordinate per iteration, and there are multiple rules for choosing the coordinate at each iteration. Here, we follow Wang et al. [2019] and simply pick the $i$-th coordinate of $\lambda$ such that its projected gradient is the largest. We describe greedy coordinate ascent in Algorithm 2. To avoid the full gradient computation at every iteration, we keep track and update it given that $\lambda$ only changes by one coordinate.

---

**Algorithm 2:** Greedy Coordinate Ascent

**Data:** Test point $(x, y)$, Voronoi cell described by $Az \leq b$
**Result:** Projection of $x$ onto the Voronoi cell
1 Initialize $\lambda \leftarrow \mathbf{0}$
2 **for** $t \in \{1, \ldots, T\}$ **do**
3     $\nabla g(\lambda) \leftarrow -AA^\top \lambda + Ax - b$
4     $j \leftarrow \arg\max_i |(\max\{\lambda + \nabla g(\lambda), 0\} - \lambda)_i|$
5     $\lambda_j \leftarrow \max\left\{\lambda_j + \frac{\nabla g(\lambda)_j}{\|a_j\|_2^2}, 0\right\}$
6 **end**
7 **return** $z = x + A^\top \lambda$;

---

### F.1 Details on Bisector Activeness Testing

We do a total of three checks to determine the feasibility: (i) check if the dual objective converges fast. When unbounded, the dual objective diverges or keeps increasing with a constant rate or faster.

Additionally, we test whether the KKT conditions hold at the end of the optimization. Namely, (ii) check if the primal residual is zero, and (iii) check if the complementary slackness is satisfied. When all three checks pass, we conclude that the bisector is active. Otherwise, it is considered inactive, and there is no need to finish the distance computation or insert it to $PQ$.