# OpenReview forum: "Adversarial Examples for k-Nearest Neighbor Classifiers Based on Higher-Order Voronoi Diagrams"
_NeurIPS.cc/2021/Conference — NeurIPS 2021 Poster_

### Official Review · Reviewer_dxry · 2021-07-13

**Rating:** 5
**Confidence:** 3

**Summary:**

This paper focuses on the problem of generating adversarial perturbations for $k$-nearest-neighbor classifiers. Focusing on the geometric nature of $k$-nearest-neighbor classification, the proposed scheme GeoAdEx works with order-$k$ Voronoi cells and facets to search around the test point being attacked by gradually increasing the search radius around it and updating the upper and lower bounds to the optimal adversarial perturbation norm. Beyond this novel algorithm to generate the optimal perturbation, the paper focuses on making this algorithm meaningfully scalable in high dimensional data with $k>1$ nearest-neighbors. The first optimization is the observation that the task of finding the neighboring order-$k$ Voronoi cells to the current cell in question can be solved by finding the neighboring order-$1$ Voronoi cells -- this is an improvement though finding order-$1$ Voronoi cells in high dimensions is still a very significant bottleneck, and hence needs to be approximated by just using the $m$ nearest neighbors of all the $k$ points in the current cell. This introduces a significant approximation that appears to be necessary to make the proposed algorithm tractable. Without this approximation, the proposed algorithm is guaranteed to find optimal perturbation; the above approximation does not have any such property. Empirically, the proposed algorithm is evaluated against existing baselines, with significant improvement, however, with an order of magnitude more computational cost. The empirical evaluation also study the effect of the class separation on the relative performance of the proposed GeoAdEx against baselines, demonstrating the intuitive result that GeoAdEx performs better with small class separations since the baselines tend to miss the closeby small order-$k$ Voronoi cells which GeoAdEx almost exhaustively searches over.


**Limitations And Societal Impact:**

The authors have discussed limitations which is mainly the computational complexity of this proposed scheme. The authors have not discussed any potential negative societal impact.


**Main Review:**

## Positives

- This paper focuses on a very interesting problem on generating adversarial perturbations for $k$-nearest-neighbor classifiers and differs from recent existing literature in that the existing literature leverages optimization based approaches combined with heuristics to handle $k > 1$ while the proposed GeoAdEx leverages a geometric breadth-first search over all order-$k$ Voronoi cells to find the optimal adversarial perturbations.
- The paper is very well-written and easy to follow, explaining most parts explicitly. The paper also provides good discussions of
  - the different aspects of the GeoAdEx algorithm,
  - the various design choices while clearly contrasting against existing work,
  - the optimizations and approximations made to make the proposed GeoAdEx computationally tractable, and
  - the results of the empirical evaluation, explicitly highlighting a possible reason behind the improved performance of GeoAdEx
- The authors connect to many really recent papers, positioning the proposed scheme well.


## Questions/concerns

My main concern is the computational cost of this method, and the lack of some form of parity between the baselines and GeoAdEx. The original GeoAdEx is essentially performing a breadth-first exhaustive search over the set of order-$k$ Voronoi cells. This makes the computational time fairly intractable. However, when comparing to baselines, it is not clear how the performance would compare when all schemes use approximately the same amount of time. The authors claim that some of the baselines cannot really perform better even if they are given more amount of time. In such cases, it would be good to understand what level of performance GeoAdEx is able to obtain within the time used by the baselines (potentially by leveraging early-stopping).

Moreover, it is not clear how robust the proposed scheme is in terms of its empirical performance:
- It is not clear how sensitive GeoAdEx is to the choice of $m$ in the approximate version and other hyperparameters.
- It would be good to understand the "convergence" of GeoAdEx in terms of the "runtime/iterations vs perturbation norm" tradeoff.
- It is not clear how the hyperparameters of the proposed scheme and the baselines were tuned.
- How critical the proposed scheme is to the Sitawarin & Wagner (2020) initialization of $\epsilon$ in terms of computational time vs. perturbation norm tradeoffs.
- The results presented in the main paper do not have any error bars or significance tests so it is not clear how significant these gains for GeoAdEx are over the baselines.


### Specific Questions:

Beyond the above concerns, there are some specific critical parts of the paper that are not clear to me. Most of these following questions pertain to me trying to understand how we can perform some of the mentioned operations in the proposed algorithm in a non-exhaustive manner:

- How is the "visited" order-$k$ Voronoi cell tracked? Do we ever need to explicitly write out all the $n\choose{k}$ order-$k$ Voronoi cells? How is a particular facet connected to a cell?
- L128-129 "GeoAdEx needs to ... with cell $\Psi$": How easy is it to enumerate the facets than comprise a cell $\Psi$? How is it done efficiently?
- L207-208 ".. we can safely filter out these facets ...": How do we do the filtering without computing distance to facet?
- L230-232: $x_{\pi(m)}$ should not be the $m$-th nearest-neighbor of $x_i$ but the $m$-th nearest-neighbor not in $\{x_1, \ldots, x_k\}$. Also, as per the definition, $\widetilde{nb}_m(x_i)$ will change for any given $x_i$ as the algorithm progresses so these $m$-neighbors have to found for each cell. Is that right? This makes the computation cost still fairly high -- how is this handled?
- L253-256: "However, the second limitation can be addressed ....": It is not clear how the second limitation can be addressed without having the exact $\cup_{i=1}^k nb(x_i)$. This needs to be clarified.
- Lemma 2 follow from Lemma 1 and then structure of the algorithm. What is the main novelty/technical difficulty here?


**Time Spent Reviewing:**

4

---

> ### Author Response · Authors · 2021-08-11
> **Author Response of Paper2529 to Reviewer dxry (2/2)**
>
> - "Also, as per the definition, $\widetilde{nb}\_m(x\_i)$ will change for any given $x_i$ as the algorithm progresses so these $m$-neighbors have to be found for each cell. Is that right? This makes the computation cost still fairly high -- how is this handled?"
>
> To find the $m$-nearest neighbors, we use the same data structure as the $k$-NN itself, which is simply a linear search. It does not need to be set up and has a search complexity of $O(n)$. Alternatively, one can use more advanced data structures such as a k-d tree, but they generally have more hyperparameters to consider and require some time to set up.
>
> - "L253-256: 'However, the second limitation can be addressed ....': It is not clear how the second limitation can be addressed without having the exact $\cup_{i=1}^k nb(x_i)$. This needs to be clarified."
>
> The second limitation is that a cell from the approximate version can be a _superset_ of the true cell. This is the case because the true cell is partitioned by _all_ of the hyperplanes, but in the approximate version, GeoAdEx may use only a _subset_ of the neighbors, and thus, the cell may be partitioned only by a subset of the hyperplanes. When an adversarial cell is larger than what it is supposed to be, it may also be (falsely) closer to the test point. This potentially leads to an incorrect upper bound. Luckily, it is simple to avoid this problem. We can simply make sure that we use all of the hyperplanes (i.e., all generators, not just the approximate neighbors) when computing distance to any adversarial cell, and thus, the distance is guaranteed to be correct. For the non-adversarial cells, we can still use the approximation for the speed-up, and we may miss some cells (the first limitation). However, it does not affect the correctness of GeoAdEx.
>
> - "Lemma 2 follows from Lemma 1 and then structure of the algorithm. What is the main novelty/technical difficulty here?"
>
> Lemma 1 and 2 are adapted from Jordan et al. [2019] where they are stated for neural networks. We repurpose them and make the necessary connections to $k$-NN classifiers and GeoAdEx in Appendix D.2 and D.3. They do not represent our main contribution but are required to guarantee the correctness of GeoAdEx. Our main contributions come from Theorem 1, the approximation in Section 3.5.2, as well as multiple techniques that make GeoAdEx scalable so that it surpasses all of the baselines.

---

> ### Author Response · Authors · 2021-08-11
> **Author Response of Paper2529 to Reviewer dxry (1/2)**
>
> We would like to thank the reviewer for the detailed and constructive feedback on our work. In the list below, we restate and then address the concerns/questions that were raised.
>
> - "However, when comparing to baselines, it is not clear how the performance would compare when all schemes use approximately the same amount of time. The authors claim that some of the baselines cannot really perform better even if they are given more amount of time. In such cases, it would be good to understand what level of performance GeoAdEx is able to obtain within the time used by the baselines (potentially by leveraging early-stopping)."
> - "It would be good to understand the 'convergence' of GeoAdEx in terms of the 'runtime/iterations vs perturbation norm' tradeoff."
>
> We agree with the reviewer that such comparisons will strengthen the results. We ran several experiments with different hyperparameters of both GeoAdEx (Appendix B.2, Figure 3) and the baselines (Appendix B.4, Table 5) which can be used to compare the runtime. Furthermore, we conducted an additional experiment based on this comment in which we use a wider range of hyperparameters and datasets. Please find the results in this [comment](https://openreview.net/forum?id=2j3B_YkC8r&noteId=6dHf6QfaCwK) (named "Extra experiments for runtime comparison"). Both the existing and the new results suggest the same conclusion. In the regime with very short runtimes (~100s or ~1s per sample), GeoAdEx generally finds larger adversarial perturbations than the baselines. However, with a longer runtime, it outperforms them by a large margin. In most settings, the previous works cannot minimize their adversarial distance further with more computation time because of the non-exhaustive heuristics they employ. On the other hand, GeoAdEx can utilize the extra time limit and find smaller adversarial distances.
>
> - "It is not clear how sensitive GeoAdEx is to the choice of $m$ in the approximate version and other hyperparameters."
> - "It is not clear how the hyperparameters of the proposed scheme and the baselines were tuned."
>
> We describe how the hyperparameters are chosen in Appendix A. Essentially, we use the baselines' publicly available code as well as their default hyperparameters. Additionally, we also fine-tune the hyperparameters of all the baselines to see if they can find smaller adversarial perturbations. However, as mentioned above, we do not observe any significant difference, and for Yang et al. [2020] and Wang et al. [2019], doing so only leads to a longer runtime with no improvement on the distance. The results are presented in Table 5 (Appendix B.4) as well as [the new results](https://openreview.net/forum?id=2j3B_YkC8r&noteId=6dHf6QfaCwK).
>
> For GeoAdEx, we compare the hyperparameters, $m$ and time limit, in Figure 3 (Appendix B.3) and [the new results](https://openreview.net/forum?id=2j3B_YkC8r&noteId=6dHf6QfaCwK). For Table 1, we choose the time limit of 100 seconds to make sure that we stay within our computation capacity. The choice of $m = 40$ is slightly arbitrary. We simply try several values of $m \in \\{ 10, 20, 40, 60 \\}$ but do not find any significant difference. We deliberately do not fine-tune our hyperparameters and use the same ones for all datasets and values of $k$ for fair comparisons.
>
> - "How critical the proposed scheme is to the Sitawarin & Wagner (2020) initialization of $\epsilon$ in terms of computational time vs. perturbation norm tradeoffs."
>
> We conduct new experiments, based on this comment, where we skipped the initialization scheme and simply set $\epsilon$ to $\infty$. We observe that GeoAdEx increases the computation time by 25% (13%), 107% (92%), and 14% (35%) for Australian, Covtype, Diabetes, respectively with $k=3$ ($k=5$). We will make sure to include the comparison for all of the datasets and values of $k$ in the final version of the paper.
>
> - "The results presented in the main paper do not have any error bars or significance tests so it is not clear how significant these gains for GeoAdEx are over the baselines."
>
> We are not sure if we understand this comment. Please feel free to elaborate if we got the wrong idea. All the algorithms in this study, including GeoAdEx, are deterministic and hence, the adversarial distance computed is also deterministic. Thus, we believe that error bars or significance tests may not be applicable. Furthermore, we make sure to run all of the algorithms on the same set of train/test data so that the results are directly comparable.
>
> - "How is the 'visited' order-$k$ Voronoi cell tracked?"
>
> We refer the reviewer to the relevant text in Section 3.1, "...we use a dictionary that has an entry for every order-$k$ Voronoi cell that the algorithm has visited." This data structure is a `set` in Python which is very similar to `dict` and is a hashtable (so checking membership is $O(1)$). The visited cells are represented by the index of their $k$ generators.
>
> - "Do we ever need to explicitly write out all the order-$k$ Voronoi cells?"
>
> We never need to explicitly list all of the order-$k$ cells, and that is the beauty of a local search based approach. GeoAdEx lists only the neighbors of the current cell in the search which are then pushed to a queue to be searched next.
>
> - "How is a particular facet connected to a cell?"
>
> We are not sure if we understand the meaning of this question.
> => From an algorithmic perspective, the priority queue essentially keeps track of the facets and their corresponding cells that GeoAdEx searched (on top of the distance and a flag whether the facet is adversarial). Similar to the visited cells, these facets are populated on the fly whenever a cell is searched by GeoAdEx.
> => For how these facets (both active and inactive) are listed out in the first place, any order-$k$ cell has $n(n-k)$ facets in total which can be simply computed by pairing each of the $k$ generators associated with this cell to any of the remaining $n-k$ generators.
>
> - "L128-129 'GeoAdEx needs to ... with cell ': How easy is it to enumerate the facets than comprise a cell? How is it done efficiently?"
>
> The answer to this question is presented in Section 3.2. As we discussed, we chose to proceed with approach (C) which avoids computing an exact enumeration of the order-$k$ neighbors or constructing the full order-$k$ Voronoi diagram. This approach considers only first-order neighbors, and we use Theorem 1 to make the connection back to the $k$th-order. This is the key that makes GeoAdEx scalable to larger values of $k$. A step that is not discussed in detail in this section is how the facet’s activeness is tested. This is briefly mentioned in Section 3.3 under "Testing Bisector Activeness," and more details are given in Appendix F.1. To summarize, we can check the activeness as we solve the QP for computing the distance. If a facet is inactive, the QP would be infeasible, and thus, we can check whether the dual problem is unbounded. We also deploy additional checks to make sure that we stop the optimization as soon as possible to save computation time.
>
> - "L207-208 '.. we can safely filter out these facets ...': How do we do the filtering without computing distance to facet?"
>
> The details of the filtering process as well as the other speed-up techniques are included in Appendix E. There are two places in the algorithm where the filter can be used.
> 1. Before the distance computation: From Appendix E.1 (L598-601), "specifically, we can compute the orthogonal projection of $x$ onto the bisector implied by the facet. If the distance to the bisector, which is a lower bound on the distance to the facet, is larger than $\epsilon$, then this facet can be safely discarded."
> 2. During the distance computation: Also from Appendix E.1 (L602-605), "even if we proceed with the distance computation, we can still use $\epsilon$ to terminate the optimization in Eqn. (2) early. Specifically, if the dual objective of Eqn. (2) surpasses $\epsilon^2$, we can terminate the solver and discard this facet since, from strong duality, the dual objective is a lower bound of the primal objective, i.e. the (squared) distance between $x$ and this facet."
>
> - "L230-232: $x_{\pi(m)}$ should not be the $m$-th nearest-neighbor of $x_i$ but the $m$-th nearest-neighbor not in $\\{x_1,\dots,x_k\\}$."
>
> This is up to the definition of $\widetilde{nb}\_m(x\_i)$ (L229). Here, we define $\widetilde{nb}\_m(x\_i)$ as the set of neighbors of $x_i$ that are closer to $x_i$ than its $m$-th nearest-neighbor but excluding $\\{x_1,\dots,x_k\\}$. In other words, $m - k \le |\widetilde{nb}\_m(x\_i)| \le m$. With this definition, $x_{\pi(m)}$ is simply the $m$-th nearest-neighbor of $x_i$ which may include $\\{x_1,\dots,x_k\\}$. On the other hand, if we want to force $|\widetilde{nb}\_m(x\_i)|$ to exactly be $m$, $x_{\pi(m)}$ will have to exclude $\\{x_1,\dots,x_k\\}$ as suggested. We believe that this choice is arbitrary and does not affect any interpretation or outcome of the algorithm since $m$ is treated as a hyperparameter.

---

> > ### Comment · Reviewer_dxry · 2021-08-25
> > **Response received**
> >
> > Thank you for the various clarifications and the pointers to the supplement that contained more details and empirical evaluations. I think many of my clarification questions have been addressed.
> >
> > I have a couple of follow-up questions/comments:
> > - Regarding "error-bars", I was referring more to different splits of the training data set and attacked test points.
> > - While the decreasing and then increasing trend of GeoAdEx with increasing time is explained, it is not clear why there are situations where increasing $m$ actually decreases runtime while also decreasing perturbations (See for example Covertype and Diabetes). This seems a little odd.
> > - Thank you for the updated results that ablate the effect of the S&W initialization. It appears that the use of the S&W initialization can result in anywhere betwee 10-100% improvement in runtime. Do we have any intuition on when we would expect significant improvements and when the improvements would be nominal.
> > - On a related note, can you clarify if the cost of the S&W optimization (for the initialization of GeoAdEx) is included in the runtime for GeoAdEx for the evaluations in the paper or the additional experiments? This just adds another dimension where we need to decide how long we should run the S&W optimization for. Do you have any intuition here?

---

> > > ### Author Response · Authors · 2021-09-01
> > > **Response to Reviewer dxry**
> > >
> > > We would like to thank Reviewer dxry for all the constructive comments and questions. We will be sure to include the additional experiments and observations made during this discussion in the final version of the paper.
> > >
> > > - Regarding "error-bars", I was referring more to different splits of the training data set and attacked test points.
> > >
> > > Thank you for clarifying your question. We agree that the confidence intervals would benefit the interpretation of the improvement. However, computing confidence intervals across data partitions for all our experiments in Table 1 is very expensive. As a result, we decided to fix the random seed to the same one used by Yang et al. for consistency (e.g., the same training and test samples). Based on our rough calculation, it will take approximately 30 days of an uninterrupted run on our server to compute confidence intervals with $N = 10$. Since our machine is shared between multiple colleagues, it most likely will not finish before the rebuttal deadline, but we will make sure to include it in the final version of the paper. As of now, we will try to run as many as possible and update you on this thread.
> > >
> > > - While the decreasing and then increasing trend of GeoAdEx with increasing time is explained, it is not clear why there are situations where increasing $m$ actually decreases runtime while also decreasing perturbations (See for example Covertype and Diabetes). This seems a little odd.
> > >
> > > Thank you for bringing this up. We also noticed this interesting trend and came up with the following hypothesis to explain it for different values of $m$, namely the small-$m$ and the large-$m$ regions. Additionally, we conducted new experiments to verify our hypothesis.
> > >
> > > **Small-$m$ region.** When a smaller $m$ is used with GeoAdEx, fewer first-order neighbors are considered, and thus, the search has a higher chance of missing facets and nearby adversarial cells completely. As a result, it has to expand the search radius which, in turn, would discover adversarial examples that are further away and use a longer runtime. Conversely, if we increase $m$, we may find these previously missed cells and terminate earlier, resulting in both lower adversarial distance and runtime. We call the first scenario from the above exposition, the “small-$m$ region.”
> > >
> > > **Large-$m$ region.** On the other hand, when $m$ is sufficiently large and no adversarial cell is missed, increasing $m$ could have a reversed effect. In particular, when $m$ increases, more first-order neighbors have to be considered, and hence more _nearby_ cells will have to be searched. For each of the test samples, this could lead to (i) an increased runtime and/or (ii) the previously found adversarial cells that are further away may now be missed instead. When (ii) happens, GeoAdEx will timeout and just return the initialized upper bound. Therefore, in the “large-$m$ region,” both the adversarial distance and the total runtime may increase with $m$.
> > >
> > > We verify this hypothesis by inspecting the number of samples that are timed out by GeoAdEx. If our hypothesis holds, when we test different values of $m$, we expect to see a decreasing trend on the number of timeouts in the small-$m$ region and an increasing trend in the large-$m$ region. Specifically, when varying $m \in \\{5,10,20,40,60,80,100,120\\}$ on the Diabetes dataset, we observe the following number of timeouts over 100 samples: 41, 21, 14, 16, 10, 15, 16, and 17. The first three experiments ($m=5,10,20$) which correspond to the small-$m$ region show a decreasing number of timeouts from 41 to 14. The last three experiments ($m=80,100,120$) correspond to the large-$m$ region where both runtime and distance increase with $m$. The same phenomenon also happens on Covtype given the same hyperparameters and the range of values of $m$. In this case, the numbers of timeouts are 24, 9, 3, 3, 3, 4, 4, and 5, respectively.
> > >
> > > Note that whether a value of $m$ is considered “small” or “large” varies by datasets and the time limit. Increasing the time limit reduces the number of timeouts and hence delays the large-$m$ region (i.e., occurs at a larger $m$). Additionally, this observation leads to two practical suggestions: (1) regardless of $m$, increasing the time limit is always beneficial in terms of the adversarial distance, but (2) for a fixed time limit, there is an optimal value of $m$.
> > >
> > > - Thank you for the updated results that ablate the effect of the S&W initialization. It appears that the use of the S&W initialization can result in anywhere between 10-100% improvement in runtime. Do we have any intuition on when we would expect significant improvements and when the improvements would be nominal.
> > >
> > > While effects of the initialization are generally minor, we found two cases, namely Covtype and Fourclass, where the initialization leads to a large improvement in the runtime. Without the initialization, the runtime increases by 107% and 300% in these two datasets respectively. See [this plot](https://imgur.com/xKC1hzm) for the other datasets and for the absolute runtime. We give our explanation for these two datasets below:
> > > For Covtype, GeoAdEx’s runtimes are 561s with the initialization step and 1163s without. Without an initial upper bound on $\epsilon^*$, there is a small chance that GeoAdEx misses a nearby adversarial cell and keeps running until the time limit is met. While only several test samples are affected, it ends up raising the total runtime by a relatively large margin (100% from ~500s) because the time limit is set to 100s per sample.
> > > The initialization has the same kind of effect on Fourclass as on Covtype, but it affects a much larger number of samples. The number of timeouts goes from 4 to 43 as the initialization is removed, explaining the significant increase in the total runtime. One hypothesis is that this phenomenon takes place because S&W attack is particularly effective on Fourclass (note its competitive performance compared to the other attacks), and thus removing it as an initialization results in an equally large degradation on the performance of GeoAdEx.
> > >
> > > - On a related note, can you clarify if the cost of the S&W optimization (for the initialization of GeoAdEx) is included in the runtime for GeoAdEx for the evaluations in the paper or the additional experiments? This just adds another dimension where we need to decide how long we should run the S&W optimization for. Do you have any intuition here?
> > >
> > > Yes, the cost of S&W initialization is included in GeoAdEx runtime. Deciding how long to run S&W initialization is a good question from a practical standpoint. Originally, we just wanted any upper bound on the optimal adversarial distance $\epsilon^*$ as long as it is approximately within the same order of magnitude. As a result, we simply choose S&W attack because it finds an adversarial example quickly (even though the upper bound is relatively loose) with little impact on the total runtime. The choice is somewhat arbitrary and can be replaced with the attacks by Yang et al. or Wang et al.
> > >
> > > Determining how long to run the initialization attack depends on many factors including the dataset, the desired time limit, and hyperparameters of GeoAdEx. So it should be considered on a case-by-case basis, perhaps by a hyperparameter optimization on a small subset of the data. A cheaper strategy that works across all datasets is to terminate the attack (per-sample) when the adversarial distance starts to plateau (e.g, less than A% improvement in the last B iterations).
> > >
> > > To test the effect of the initialization’s hyperparameters on GeoAdEx, we ran a simple experiment on the Diabetes dataset by varying the runtime of S&W initialization and the time limit of GeoAdEx. We plot the total runtime vs. the adversarial distance [here](https://imgur.com/nF7pyKr). Each curve uses a different scaling factor of the S&W attack as described in our previous experiment to vary its runtime. Each dot in the curve is generated by varying the time limit of GeoAdEx from 5 to 200 seconds. For a wide range of total runtimes, S&W initialization with a scaling factor between 2 and 3 (or anywhere between 4% and 35% of the total runtime) seems like an optimal spot on the runtime and adversarial distance trade-off. This suggests that the performance of GeoAdEx is fairly insensitive to the hyperparameters of the S&W initialization. Based on this observation alone, one may aim to run the initialization for ~10-20% of the total runtime as the simplest baseline and avoid any additional overhead.

---

> > > > ### Author Response · Authors · 2021-09-17
> > > > **Response to Reviewer dxry (results with error bars)**
> > > >
> > > > The tables below display the mean adversarial distance and the runtime with error bars (10 repeated runs with different train-test data shufflings to compute confidence intervals) for $k=3$. We try to “tune” hyperparameters of the baselines so that their runtimes are approximately less than one magnitude away from the runtime of GeoAdEx. This reduces the runtime discrepancy between GeoAdEx and the baselines in the original version of Table 1. Specifically, we scaled hyperparameters of S&W and Wang et al. by a factor of 4 in the same way that we previously explained. We also make sure that all of the attacks run on the same set of hyperparameters for all datasets to avoid any fine-tuning. Consequently, we did not change the hyperparameters of Yang et al. since it already takes a long time on some datasets.
> > > >
> > > > This experiment makes the main table more informative but does not change the trend or the conclusion we previously made. However, it should not be taken as a thorough runtime comparison which is already shown in the first post of this forum and will be included in the final version of the paper. We will continue to run the experiments for $k=5,7$ and update this thread.
> > > >
> > > >
> > > > ### Mean adversarial distance
> > > >
> > > > | $k$ | Attacks            |          Australian |             Covtype |            Diabetes |           Fourclass |            Gaussian |             Letters |          F-MNIST-06 |
> > > > | --- | ------------------ | ------------------: | ------------------: | ------------------: | ------------------: | ------------------: | ------------------: | ------------------: |
> > > > | 3   | S&W [2020]         |   $0.4242\pm0.0201$ |   $0.1931\pm0.0148$ |   $0.1055\pm0.0068$ |   $0.1079\pm0.0039$ |   $0.0442\pm0.0028$ |   $0.1128\pm0.0049$ |   $0.1554\pm0.0121$ |
> > > > |     | Yang et al. [2020] | $0.4658 \pm 0.0258$ | $0.2385 \pm 0.0101$ | $0.1392 \pm 0.0077$ | $0.1209 \pm 0.0055$ | $0.1138 \pm 0.0023$ | $0.1370 \pm 0.0046$ | $0.1773 \pm 0.0043$ |
> > > > |     | Wang et al. [2019] | $0.4466 \pm 0.0205$ | $0.2134 \pm 0.0170$ | $0.1164 \pm 0.0052$ | $0.1117 \pm 0.0039$ | $0.0825 \pm 0.0030$ | $0.1218 \pm 0.0050$ | $0.1643 \pm 0.0047$ |
> > > > |     | GeoAdEx            | $0.3610 \pm 0.0184$ | $0.1450 \pm 0.0118$ | $0.0786 \pm 0.0060$ | $0.1114 \pm 0.0060$ | $0.0431 \pm 0.0025$ | $0.1092 \pm 0.0060$ | $0.1579 \pm 0.0106$ |
> > > > |     |                    |            $0.8510$ |            $0.7509$ |            $0.7450$ |            $1.0324$ |            $0.9751$ |            $0.9681$ |            $1.0161$ |
> > > >
> > > > - For all numbers, lower is better.
> > > > - The reported errors are 95%-confidence interval according to Student's $t$-distribution with $10$ samples.
> > > > - The last row reports the mean adversarial distance of GeoAdEx divided by the smallest distance of the three baselines.
> > > >
> > > > ### Runtime in seconds
> > > >
> > > > | $k$ | Attacks            |     Australian |        Covtype |       Diabetes |      Fourclass |       Gaussian |        Letters |     F-MNIST-06 |
> > > > | --- | ------------------ | -------------: | -------------: | -------------: | -------------: | -------------: | -------------: | -------------: |
> > > > | 3   | S&W [2020]         |    $654 \pm 1$ | $1225 \pm 250$ |  $465 \pm 121$ |    $336 \pm 5$ |   $811 \pm 14$ |  $3372 \pm 13$ |  $972 \pm 179$ |
> > > > |     | Yang et al. [2020] |      $8 \pm 1$ |   $925 \pm 97$ |      $9 \pm 1$ |      $7 \pm 1$ |    $599 \pm 6$ |  $1555 \pm 38$ |  $2151 \pm 62$ |
> > > > |     | Wang et al. [2019] |   $361 \pm 21$ |   $363 \pm 20$ |    $128 \pm 9$ |    $129 \pm 5$ |   $226 \pm 19$ |   $259 \pm 19$ |   $272 \pm 21$ |
> > > > |     | GeoAdEx            | $3284 \pm 362$ |  $560 \pm 134$ | $1947 \pm 311$ | $1572 \pm 123$ | $2509 \pm 387$ | $3800 \pm 300$ | $6210 \pm 262$ |

---

### Official Review · Reviewer_FLts · 2021-07-14

**Rating:** 7
**Confidence:** 3

**Summary:**

This paper proposed a method to run adversarial attacks on a k-NN classifier based on a breadth-first search. The proposed algorithm gradually expands the radius of searching until an adversarial example is found. The authors theoretically studied the optimality of the proposed method and proposed necessary approximations. The limitation and advantages of this method are also addressed. Finally, the experimental results show the effectiveness of this method.

**Limitations And Societal Impact:**

The authors adequately addressed the limitations and potential negative societal impact of their work

**Main Review:**

In general, I think this paper is well-written and intuitive. The theoretical studies are rigorous. I briefly checked the proof and they seem to be correct to me. Though the running time still poses the biggest disadvantage of this method, the experimental results show clear improvement to the existing method in terms of the mean norm. So I am recommending an accept. My detailed comments are as follows.

> I think more explanation is needed for the approximation proposed in 3.5.2, especially how fast would this approximation converge to the original problem. As mentioned in line 244, this has a domino effect on the result of the adversarial distance, I think it would be great if the authors can provide a more detailed analysis of this approximation.

> Given the dramatic increase in running time in Table 4, I think it may be necessary to also have a figure showing the proposed method's performance vs. running time by terminating it at different times.

> Minor: please refrain from only using color to distinguish curves as in Figure 3 as it is not friendly to readers with color-blindness.

**Time Spent Reviewing:**

3

---

> ### Author Response · Authors · 2021-08-11
> **Author Response of Paper2529 to Reviewer FLts**
>
> We would like to thank the reviewer for the detailed and constructive feedback on our work. In the list below, we restate and then address the concerns/questions that were raised.
>
> - "I think more explanation is needed for the approximation proposed in 3.5.2, especially how fast would this approximation converge to the original problem. As mentioned in line 244, this has a domino effect on the result of the adversarial distance, I think it would be great if the authors can provide a more detailed analysis of this approximation."
>
> More rigorous and formal analyses on the approximation error are indeed interesting to study but also challenging to quantify for the following reasons. First, we need to gain a better understanding of $k$-NN for arbitrary dimensions and larger $k$, which is not trivial based on the state-of-the-art results in computational geometry. Second, it heavily depends on the data distribution which would require strong assumptions to work with. We agree that it is an important direction for future work along with exploring better approximations.
>
> - "Given the dramatic increase in running time in Table 4, I think it may be necessary to also have a figure showing the proposed method's performance vs. running time by terminating it at different times."
>
> We agree with the reviewer that such comparisons will strengthen the results. We ran several experiments with different hyperparameters of both GeoAdEx (Appendix B.2, Figure 3) and the baselines (Appendix B.4, Table 5) which can be used to compare the runtime. Furthermore, we conducted an additional experiment based on this comment in which we use a wider range of hyperparameters and datasets. Please find the results in this [comment](https://openreview.net/forum?id=2j3B_YkC8r&noteId=6dHf6QfaCwK) (named "Extra experiments for runtime comparison"). Both the existing and the new results suggest the same conclusion. In the regime with very short runtimes (~100s or ~1s per sample), GeoAdEx generally finds larger adversarial perturbations than the baselines. However, with a longer runtime, it outperforms them by a large margin. In most settings, the previous works cannot minimize their adversarial distance further with more computation time because of the non-exhaustive heuristics they employ. On the other hand, GeoAdEx can utilize the extra time limit and find smaller adversarial distances.
>
> - "Minor: please refrain from only using color to distinguish curves as in Figure 3 as it is not friendly to readers with color-blindness."
>
> Regarding Figure 3, we apologize for this mistake, and we will make sure to use additional indicators in the final version.

---

> > ### Comment · Reviewer_FLts · 2021-08-24
> > **Post rebuttal**
> >
> > I appreciate the authors' feedback. My concerns are addressed. I'll keep my recommendation for an accept and let the ac to decide.

---

> > > ### Author Response · Authors · 2021-09-01
> > > **Response to Reviewer FLts**
> > >
> > > We are glad to hear that our response has addressed your concerns. We would like to thank the reviewer again for the time and effort spent in the review process. We truly appreciate all the questions and the constructive feedback which will help improve this paper in its next iteration.

---

### Official Review · Reviewer_hZxG · 2021-07-15

**Rating:** 7
**Confidence:** 3

**Summary:**

Prior adversarial attacks on k-nearest neighbor (k-NN) do not scale well with the increase of k and the number of data. This paper proposed a set of better heuristics towards speeding up this problem. Theoretically, they show that their method still finds the optimal solution if no time limit. Empirically, they show that their methods can provide stronger attacks than prior works.

**Limitations And Societal Impact:**

The limitations are described clearly on page 7, and I am not aware of any potential negative social impact of this paper.


**Main Review:**

Pros:
- Written clearly.
- Code is provided in the supplementary.
- The proposed method seems to be a nice improvement over the prior work in terms of speed.

Cons:
- In one of your cited papers (Yang et al., [2020]), they also proposed a defense algorithm proposed for $k$-NN, how does your algorithm work on $k$-NN with the defense applied?

minor:
- It is a bit confusing for Table 4 in the supplementary material and the text around it. It appears that Yang et al. is faster in many of the datasets (Australian, Covtype, Diabetes, and Fourclasses) than GeoAdEx. But the description says generally GeoAdEx is faster. If I am reading it correctly, the authors should mean that GeoAdEx is faster when the dataset and k are larger (instead of ``in general'').
- Can the method proposed in this work be applied to the certification of relu networks (the setting in Jordan et al. [2019] cited in the paper)? Can it perform well there?

**Time Spent Reviewing:**

3

---

> ### Author Response · Authors · 2021-08-11
> **Author Response of Paper2529 to Reviewer hZxG**
>
> We would like to thank the reviewer for the detailed and constructive feedback on our work. In the list below, we address the concerns and the questions that were raised.
>
> - "It is a bit confusing for Table 4 in the supplementary material and the text around it. It appears that Yang et al. is faster in many of the datasets (Australian, Covtype, Diabetes, and Fourclasses) than GeoAdEx. But the description says generally GeoAdEx is faster. If I am reading it correctly, the authors should mean that GeoAdEx is faster when the dataset and k are larger (instead of 'in general')."
>
> Yes, your understanding is correct. We agree that the texts around Table 4 could have been clearer. In the next version of our paper, we will clarify the comparison per dataset and value of $k$.
>
> - "Can the method proposed in this work be applied to the certification of relu networks (the setting in Jordan et al. [2019] cited in the paper)? Can it perform well there?"
>
> The techniques we use (Theorem 1 and approximation in Section 3.5.2) cannot be directly translated to piecewise linear neural networks because "cells" for $k$-NN models have a different geometric meaning compared to cells in neural networks. Specifically, cells in a Voronoi diagram are constructed around existing generators (which we utilize in GeoAdEx) while cells in neural networks are not. However, we are the first to deploy efficient approximations/heuristics that speed up the computation, and this is indeed an interesting extension that can be applied (with the proper adjustments) to neural networks. Similar geometric techniques for neural networks (e.g., Jordan et al. [2019]) have very poor scalability, and so potentially, a different approximation could scale these techniques to more realistic models.

---

### Official Review · Reviewer_jVor · 2021-07-16

**Rating:** 6
**Confidence:** 4

**Summary:**

The paper describes a "white box" adversarial attack on k-NN classifiers that is based on the geometry of the Voronoi diagram. The paper first defines the relevant geometric elements - Voronoi cells and their facets (which reside on bisectors - hyperplanes that are equidistant to two data points) and when they are considered *adversarial*. The paper then proposes an algorithm (GeoAdEx) for searching for the nearest adversarial facet for a given test point, using a priority queue in a BFS manner. GeoAdEx is shown to provide an optimal (min L2) attack. To allow scaling GeoAdEx to k>1 and large data, several approximations are suggested (which may result in a suboptimal attack). The experimental section compares the mean attack size for seven datasets against three other kNN attacks (for k=3, 5, 7) and shows that GeoAdEx results in a smaller-norm attack.

**Limitations And Societal Impact:**

The authors discuss limitations of the proposed method. There is no negative social impact I am aware of.

**Main Review:**

Originality: The very active research area of *adversarial examples* seems to focus on deep neural networks and there are only few works discussing adversarial examples for k-NN classifiers. The current work handles this case and improves upon an earlier work, which is more heuristics-based. The presented geometric algorithm seems to be novel.

Clarity and quality: The paper is clearly written and describes the geometric setting in an accurate and easy to follow manner. In addition, the analysis of the proposed algorithm seems sound. The experimental section seems to support the claim that GeoAdEx results in smaller perturbations than earlier methods (20%-30% reduction in L2 norm). The run-time of the different attacks is not provided, but is later discussed in section 5 (being up to two orders of magnitude slower).

I am less confident about the significance of the presented work, due to several reasons:
- The basic GeoAdEx algorithm is elegant and well-justified, but in order to scale it to typical use cases, several heuristics and approximations are added. In the end, it results in a slightly better attack but at the expense of much larger runtime.
- Considering the contribution of current work to *adversarial examples* research in general (which aims at understanding why adversarial examples exist, what kind of models are vulnerable to such examples and how to make the models more robust), I'm not sure that a reduced-norm attack on a kNN classifier is a significant contribution.
- Related to the above point, the authors cite Wang et al showing that "a k-NN classifier can be as robust as the optimal Bayes classifier". It is not clear if the attack L2 values reported in Table 1 are "insignificant perturbations", similar to adversarial attacks on CNNs (indicating that kNN is not robust), or large perturbations that are required to fool a robust model. If the latter is true, the setting, in my opinion, is less interesting. It is also not clear if in some settings, a kNN classifier would be robust against earlier attacks, but vulnerable to GeoAdEx. (This is perhaps discussed in the class closeness case).


**Time Spent Reviewing:**

6

---

> ### Author Response · Authors · 2021-08-11
> **Author Response of Paper2529 to Reviewer jVor**
>
> We would like to thank the reviewer for the detailed and constructive feedback on our work. In the list below, we address the concerns and the questions that were raised.
>
> - "in order to scale it to typical use cases, several heuristics and approximations are added. In the end, it results in a slightly better attack but at the expense of much larger runtime"
>
> The default hyperparameters we choose for GeoAdEx to report in Table 1 and 4 yield a longer runtime as mentioned, but they do not capture the efficiency of GeoAdEx. For better runtime comparisons, we conducted an additional experiment that iterates through different hyperparameters of both GeoAdEx (Appendix B.2, Figure 3) and the baselines (Appendix B.4, Table 5). Additionally, to make this comparison more comprehensive, we run new experiments with a wider range of hyperparameters and datasets. The results are presented in this [comment](https://openreview.net/forum?id=2j3B_YkC8r&noteId=6dHf6QfaCwK) (named "Extra experiments for runtime comparison"). Both the existing and the new results suggest the same conclusion. In the regime with very short runtimes (~100s or ~1s per sample), GeoAdEx generally finds larger adversarial perturbations than the baselines. However, with a longer runtime, it outperforms the competition by a large margin. In most settings, the previous works cannot minimize their adversarial distance further with more computation time because of the non-exhaustive heuristics they employ. On the other hand, GeoAdEx can utilize the extra time limit and find smaller adversarial distances.
>
> - "Considering the contribution of current work to adversarial examples research in general (which aims at understanding why adversarial examples exist, what kind of models are vulnerable to such examples and how to make the models more robust), I'm not sure that a reduced-norm attack on a kNN classifier is a significant contribution."
>
> $k$-NN is one of the most well-known and simplest models studied in machine learning. We believe that having an accurate tool for evaluating its robustness holds great value for the community. It is widely used in the industry in applications such as data mining, recommendation systems, and anomaly detection systems where interpretability and simplicity are preferred [Wu et al. 2008]. $k$-NN is also an active area of research, having a lot of activities in popular libraries like Google’s ScaNN and Facebook’s FAISS. Additionally, there are a significant number of papers that utilize $k$-NN as a component of neural networks for improved robustness, e.g., Papernot and McDaniel [2018], Dubey et al. [2019], Sitawarin and Wagner [2019]. Any attacks on $k$-NN classifiers could be transferred to the corresponding defenses.
>
>
> - "Related to the above point, the authors cite Wang et al showing that 'a k-NN classifier can be as robust as the optimal Bayes classifier'. It is not clear if the attack L2 values reported in Table 1 are 'insignificant perturbations', similar to adversarial attacks on CNNs (indicating that kNN is not robust), or large perturbations that are required to fool a robust model. If the latter is true, the setting, in my opinion, is less interesting."
>
> We believe that it is crucial to have an efficient and accurate tool for evaluating the robustness of all machine learning models. As mentioned in the previous point, $k$-NN is widely used due to its simplicity and interpretability, making the robustness measurement even more important. Historically, the notion of "insignificant" or "imperceptible" perturbation is used in the adversarial example literature because experiments are conducted for neural networks and image datasets. A small $\ell_p$-norm may be a good proxy for imperceptibility for images, but in general, this depends heavily on the context and the application scenario. Defining what insignificant means in different contexts is a task that is outside of the scope of this work. $k$-NN also often operates on extracted features from raw data. A large perturbation in the feature space does not necessarily imply a large perturbation in the input space. Therefore, it should not be taken as an indicator of the perceptibility or practicality of the attack.
>
> - "It is also not clear if in some settings, a kNN classifier would be robust against earlier attacks, but vulnerable to GeoAdEx. (This is perhaps discussed in the class closeness case)."
>
> We are not sure if we understand the question correctly. We believe that the aforementioned settings are equivalent to the ones where GeoAdEx performs better than the baselines. The settings in which GeoAdEx shows significant improvement include Covtype, Diabetes, Letters, and Gaussian datasets for all values of $k = 3,5,7$. Please feel free to elaborate on the question if we misinterpreted your comment.
>
>
> References
> - Wu et al., _Top 10 Algorithms in Data Mining_, 2008.
> - Papernot and McDaniel, _Deep k-Nearest Neighbors: Towards Confident, Interpretable and Robust Deep Learning_, 2018.
> - Dubey et al., _Defense Against Adversarial Images using Web-Scale Nearest-Neighbor Search_, 2019.
> - Sitawarin and Wagner, _Defending against Adversarial Examples with k-Nearest Neighbor_, 2019.

---

> > ### Comment · Reviewer_jVor · 2021-08-25
> > **Response to Authors**
> >
> > I want to thank the Authors for the detailed explanations and clarification. After reading the response and the other reviews, I acknowledge that the importance of analyzing and improving adversarial attacks on kNN classifiers is indeed higher than I initially thought. I therefore raise my rating of the paper.

---

> > > ### Author Response · Authors · 2021-09-01
> > > **Response to Reviewer jVor**
> > >
> > > We would like to thank the reviewer again for the time and effort spent in the review process. We truly appreciate all the questions and the constructive feedback which will help improve this paper in its next iteration.

---

### Author Response · Authors · 2021-08-11
**Extra experiments for runtime comparison**

# Figures for mean adversarial distance vs. runtime

Please follow this link to view the figures: [link](https://imgur.com/a/EttJ0mn).

**Description:** We iterated through different hyperparameters of GeoAdEx as well as the baselines and plot the mean adversarial distance vs. the total runtime for each experiment which corresponds to each data point in the figure. To generate the data points for each algorithm, we started with the default hyperparameters and adjusted them (either linearly or exponentially) in the direction that should find smaller adversarial perturbations while using a longer runtime.

**Detailed Explanation:** We ran this experiment for the first three datasets in Table 1: Australian, Covtype, and Diabetes. For Yang et al. [2020], we progressively doubled the number of regions searched which is the only adjustable hyperparameter. For Wang et al. [2019], we progressively doubled the number of trials (both min and max) and the number of neighbors to consider (until it exceeds the number of all generators). There are four hyperparameters for S&W [2020]: `binary_search_steps`, `max_iterations`, `thres_steps`, and `check_adv_steps`. We progressively increased the first two and decreased the last two linearly. For GeoAdEx, we tested a more detailed breakdown by varying both $m$ (from 5 to 120) and time limit (tl). The raw data are also included in the tables below.

**How to read the plot:** This plot allows comparisons between the algorithms given a similar runtime. One way to read the plot is to fix a particular runtime and compare the mean adversarial distance from each line.

**Summary:** For Australian and Diabetes, in a regime with very short runtime (~100s total or ~1s per sample), there is at least one baseline that is both faster and finds smaller adversarial perturbation than GeoAdEx. However, with longer runtimes, our GeoAdEx outperforms all the baselines by a large margin (10%, 15% and 20% improvement over the second best for Australian, Covtype, and Diabetes, respectively). In most settings, the baselines do not benefit much, if at all, from the increased runtime. Conversely, GeoAdEx always finds smaller adversarial distances given a longer time limit.

**Additional Observations:**
- Why do GeoAdEx plots have a downward and then an upward trend in Australian and Diabetes? Each line of GeoAdEx is generated by increasing $m$ but fixing the time limit. Increasing $m$ reduces the chance of missing the adversarial facets (downward trend), but it also increases the computation time for each cell which means that there are fewer cells it can search given a fixed time limit (upward trend). This implies that there is an optimal value of $m$ for a given time limit.
- For Australian and Diabetes, the lines associated with Yang et al. [2020] are shorter than the rest because we cannot increase the total runtime by adjusting the hyperparameter any further. This is a fundamental flaw of the heuristic used by Yang et al. [2020] which only searches the cells that contain any generator from a wrong class. So the total number of cells searched is upper bound by the number of generators from a wrong class which is very limited compared to the total number of cells.
- The runtime can be slightly different from what reported in the paper since we no longer have access to the original machine. To make this figure, we run all the experiments on a server with 40 cores of Intel(R) Xeon(R) Gold 6230 CPU @ 2.10GHz.


---

# Raw Data

## Australian

### GeoAdEx (ours)

| tl \ $m$ |      5 |     10 |     20 |     40 |     60 |     80 |    100 |    120 |
| -------- | -----: | -----: | -----: | -----: | -----: | -----: | -----: | -----: |
| 2        | 0.4579 | 0.4457 | 0.4437 | 0.4372 | 0.4428 | 0.4487 | 0.4531 | 0.4534 |
|          |    146 |    187 |    197 |    204 |    213 |    222 |    227 |    233 |
| 10       | 0.4510 | 0.4308 | 0.4182 | 0.4125 | 0.4164 | 0.4204 | 0.4264 | 0.4271 |
|          |    362 |    592 |    731 |    760 |    777 |    802 |    816 |    829 |
| 50       | 0.4476 | 0.4210 | 0.4032 | 0.3976 | 0.3959 | 0.3984 | 0.3997 | 0.4018 |
|          |    693 |   1539 |   2405 |   2664 |   2563 |   2610 |   2774 |   2853 |
| 100      | 0.4467 | 0.4195 | 0.3983 | 0.3927 | 0.3945 | 0.3921 | 0.4055 | 0.4058 |
|          |    762 |   1988 |   3684 |   4318 |   3778 |   3877 |   6175 |   6334 |

"tl" = time limit per sample (seconds)
For each tl, the first row is the mean adversarial distance, and the second row is the total runtime.

### S&W [2020]

| Multiplier      |      1 |      2 |      3 |      4 |      5 |      6 |      8 |
| --------------- | -----: | -----: | -----: | -----: | -----: | -----: | -----: |
| Mean adv. dist. | 0.4910 | 0.4616 | 0.4484 | 0.4456 | 0.4451 | 0.4361 | 0.4356 |
| Runtime         |     17 |     97 |    285 |    600 |   1145 |   1791 |   3893 |

### Yang et al. [2020]

| Num. regions    |     10 |     25 |     50 |    100 |    200 |    400 |    800 |   1600 |
| --------------- | -----: | -----: | -----: | -----: | -----: | -----: | -----: | -----: |
| Mean adv. dist. | 0.4431 | 0.4428 | 0.4428 | 0.4428 | 0.4428 | 0.4428 | 0.4428 | 0.4428 |
| Runtime         |     10 |     12 |     15 |     23 |     36 |     43 |     42 |     43 |

### Wang et al. [2019]

| Multiplier      |      1 |      2 |      4 |      8 |     16 |     32 |     64 |    128 |
| --------------- | -----: | -----: | -----: | -----: | -----: | -----: | -----: | -----: |
| Mean adv. dist. | 0.4560 | 0.4527 | 0.4527 | 0.4527 | 0.4527 | 0.4527 | 0.4527 | 0.4527 |
| Runtime         |     45 |     98 |    227 |    542 |   1294 |   2197 |   2325 |   3292 |

## Covtype

### GeoAdEx (ours)

| tl \ $m$ |      5 |     10 |     20 |     40 |     60 |     80 |    100 |    120 |
| -------- | -----: | -----: | -----: | -----: | -----: | -----: | -----: | -----: |
| 2        | 0.1559 | 0.1470 | 0.1433 | 0.1441 | 0.1441 | 0.1439 | 0.1440 | 0.1444 |
|          |     93 |     97 |    102 |    110 |    115 |    120 |    124 |    129 |
| 10       | 0.1518 | 0.1419 | 0.1415 | 0.1412 | 0.1421 | 0.1403 | 0.1409 | 0.1422 |
|          |    145 |    157 |    165 |    196 |    208 |    228 |    236 |    242 |
| 50       | 0.1481 | 0.1418 | 0.1387 | 0.1385 | 0.1372 | 0.1367 | 0.1367 | 0.1367 |
|          |    323 |    358 |    299 |    389 |    432 |    524 |    551 |    620 |
| 100      | 0.1472 | 0.1390 | 0.1387 | 0.1385 | 0.1369 | 0.1365 | 0.1365 | 0.1365 |
|          |    467 |    541 |    406 |    512 |    616 |    777 |    824 |    864 |

### S&W [2020]

| Multiplier      |      1 |      2 |      3 |      4 |      5 |      6 |
| --------------- | -----: | -----: | -----: | -----: | -----: | -----: |
| Mean adv. dist. | 0.1843 | 0.1694 | 0.1614 | 0.1616 | 0.1626 | 0.1600 |
| Runtime         |     53 |    364 |   1185 |   2639 |   5145 |   8100 |

### Yang et al. [2020]

| Num. regions    |     10 |     25 |     50 |    100 |
| --------------- | -----: | -----: | -----: | -----: |
| Mean adv. dist. | 0.2881 | 0.2875 | 0.2875 | 0.2875 |
| Runtime         |    275 |    629 |   1231 |   2463 |

### Wang et al. [2019]

| Multiplier      | 1      | 2      | 4      | 8      | 16     | 32     |
| --------------- | ------ | ------ | ------ | ------ | ------ | ------ |
| Mean adv. dist. | 0.1845 | 0.1844 | 0.1844 | 0.1844 | 0.1844 | 0.1844 |
| Runtime         | 83     | 209    | 836    | 706    | 1921   | 4565   |

## Diabetes

### GeoAdEx (ours)

| tl \ $m$ |      5 |     10 |     20 |     40 |     60 |     80 |    100 |    120 |
| -------- | -----: | -----: | -----: | -----: | -----: | -----: | -----: | -----: |
| 2        | 0.1030 | 0.1017 | 0.0982 | 0.0987 | 0.1009 | 0.1015 | 0.1018 | 0.1023 |
|          |    169 |    169 |    162 |    171 |    178 |    181 |    188 |    198 |
| 10       | 0.1025 | 0.0937 | 0.0900 | 0.0899 | 0.0913 | 0.0925 | 0.0918 | 0.0918 |
|          |    486 |    471 |    472 |    491 |    525 |    549 |    567 |    575 |
| 50       | 0.0960 | 0.0876 | 0.0821 | 0.0844 | 0.0825 | 0.0823 | 0.0839 | 0.0835 |
|          |   1610 |   1381 |   1217 |   1588 |   1399 |   1521 |   1629 |   1654 |
| 100      | 0.0939 | 0.0845 | 0.0790 | 0.0802 | 0.0786 | 0.0784 | 0.0797 | 0.0802 |
|          |   2624 |   2184 |   1965 |   2474 |   2127 |   2322 |   2424 |   2524 |

### S&W [2020]

| Multiplier      | 1      | 2      | 3      | 4      | 5      | 6      | 8      |
| --------------- | ------ | ------ | ------ | ------ | ------ | ------ | ------ |
| Mean adv. dist. | 0.1080 | 0.0982 | 0.0926 | 0.0920 | 0.0929 | 0.0980 | 0.0980 |
| Runtime         | 17     | 95     | 278    | 593    | 1248   | 2042   | 3686   |

### Yang et al. [2020]

| Num. regions    |     10 |     25 |     50 |    100 |    200 |    400 |    800 |   1600 |
| --------------- | -----: | -----: | -----: | -----: | -----: | -----: | -----: | -----: |
| Mean adv. dist. | 0.1518 | 0.1516 | 0.1516 | 0.1516 | 0.1516 | 0.1516 | 0.1516 | 0.1516 |
| Runtime         |     10 |     12 |     42 |     24 |     34 |     43 |     44 |     44 |

### Wang et al. [2019]

| Multiplier      |      1 |      2 |      4 |      8 |     16 |     32 |     64 |    128 |
| --------------- | -----: | -----: | -----: | -----: | -----: | -----: | -----: | -----: |
| Mean adv. dist. | 0.1161 | 0.1154 | 0.1153 | 0.1153 | 0.1153 | 0.1153 | 0.1153 | 0.1153 |
| Runtime         |     14 |     32 |     74 |    178 |    370 |    734 |    997 |   1807 |

---

### Decision · Program_Chairs · 2021-09-27

**Decision:**

Accept (Poster)

**Comment:**

This paper studies algorithms to find adversarial examples for k-NN classifiers in a white box fashion. The reviewers have generally agreed that the work is novel, and that the experiments demonstrate an improvement over prior work. However, some comments have not been addressed in the paper, and there are many parts of the discussion that should inform improvements to the next version of the paper. For example, when designing new attack algorithms, it is important to be evaluating against good defense algorithms, such as the one from Yang et al., which appears to be missing. Similarly, it is important to provide reproducible results and error bars, especially when the improvement over existing work is small. Overall, I believe the merits outweighs the shortcoming, and that this paper provides a number of contributions to the literature around k-NN adversarial examples. I weakly recommend acceptance of this paper.